# Sparse but Critical: A Token-Level Analysis of Distributional Shifts in RLVR Fine-Tuning of LLMs

**Haoming Meng**[1]  **Kexin Huang**  **Shaohang Wei**[2]  **Chiyu Ma**[3]  **Shuo Yang**[2]
**Xue Wang**  **Guoyin Wang**[4]  **Bolin Ding**[4]  **Jingren Zhou**[4]

[1]University of Toronto   [2]Peking University   [3]Dartmouth College   [4]Alibaba Group

## Abstract

Reinforcement learning with verifiable rewards (RLVR) has significantly improved reasoning in large language models (LLMs), yet the token-level mechanisms underlying these improvements remain unclear. We present a systematic empirical study of RLVR's distributional effects organized around three main analyses: (1) token-level characterization of distributional shifts between base and RL models, (2) the impact of token-level distributional shifts on reasoning performance through cross-sampling interventions, and (3) fine-grained mechanics of these shifts at the token level. We find that RL fine-tuning induces highly sparse and targeted changes, with only a small fraction of token distributions exhibiting meaningful divergence. We further characterize the structure of these shifts through analyses of token entropy, positional concentration, and reallocation of probability mass. To assess the functional importance of these sparse changes, we conduct cross-sampling experiments that selectively swap token choices between the base and RL models. Inserting only a small fraction of RL-sampled tokens into base generations progressively recovers RL performance gains, while injecting a similarly small number of base token choices into RL-generated responses collapses performance to base levels, isolating a sparse set of token-level decisions directly responsible for RLVR's improvements. Finally, we explore divergence-weighted variants of the advantage signal as a diagnostic intervention, finding that they can yield improvements over baselines. Together, our results shed light on the distributional changes induced by RLVR and provide a fine-grained, token-level lens for understanding RLVR as a targeted refinement process.

## 1 Introduction

Recent advances in reinforcement learning with verifiable rewards (RLVR) (Lambert et al., 2024) for reasoning in large language models (LLMs), such as Group Relative Policy Optimization (GRPO) (Shao et al., 2024), have enabled substantial performance improvements on challenging reasoning and mathematical benchmarks. Despite this empirical success, the mechanisms through which RLVR modifies model behavior remain unclear.

Most evaluations of RL fine-tuning focus on aggregate metrics such as accuracy, reward, and response length. While informative, these metrics provide only a coarse-grained view of improvement and offer limited insight into *how* model behavior changes internally. In particular, a central unresolved question is: *how does RLVR reshape the token-level predictive distributions of a base model, and which of these changes actually drive downstream reasoning gains?*

Recent work has begun analyzing RL fine-tuning through token-level entropy and uncertainty perspectives (Wang et al., 2025; Cheng et al., 2025; Cui et al., 2025), highlighting the role of high-entropy tokens and exploration dynamics. However, a more detailed distributional view remains missing: how such shifts are structured across positions and contexts, how probability mass is reallocated across candidate tokens, how they evolve over training, and to what extent are they responsible for RLVR's performance gains.

In this paper, we develop a fine-grained, token-level perspective on RLVR through the lens of **distributional change**. We perform a systematic empirical study of how RLVR alters next-token distributions relative to the base model, and connect these distributional shifts directly to sequence-level reasoning performance. Our analyses reveal that RLVR acts primarily as a sparse and targeted refinement process: most token distributions remain nearly unchanged, while a small subset of high-divergence positions carries disproportionate functional importance.

Our contributions are organized as follows:

1. **Structure of Token-Level Distributional Shifts.** We show that RLVR induces sparse and highly targeted token-level distribution shifts relative to the base model. We characterize the structure of these shifts through divergence, entropy, and positional analyses, and compare across multiple methods, revealing differences in exploration and refinement behavior.

2. **Cross-Sampling Interventions.** We use forward and reverse cross-sampling interventions to measure the role of divergent token decisions. We show that modifying only a small fraction of token choices is sufficient to recover (in base-model generations) or erase (in RL-model generations) most RLVR performance gains, linking sparse distributional shifts directly to sequence-level reasoning outcomes.

3. **Fine-Grained Distribution Mechanics.** We analyze how RLVR modifies token distributions at high-divergence positions and show that it primarily reallocates probability mass within an existing candidate set rather than introducing new tokens. We support this with top-$k$ overlap, rank, tail-probability, and training-evolution analyses.

4. **Divergence-Weighted Advantage.** Motivated by these findings, we study divergence-weighted variants of the RLVR advantage signal as a diagnostic objective modification and show that they can improve over baselines.

Taken together, our results provide a unified token-level picture of RLVR fine-tuning: rather than globally rewriting model behavior, RLVR predominantly performs sparse, structured probability reallocation in a small set of critical token positions that steer downstream reasoning trajectories. This distributional and functional perspective helps clarify the mechanisms in which RLVR improves reasoning in LLMs.

## 2 TOKEN DISTRIBUTION ANALYSIS BETWEEN BASE AND RL MODELS

We begin by analyzing the general structure of distributional shifts induced by RLVR, with the goal of characterizing how token-level predictions differ between the base model and its RL-finetuned counterpart. Our analysis compares next-token distributions for both models under sequences generated by the RL policy. This framing treats the RL-generated trajectory as a reference path and allows us to quantify how the base model would need to adapt in order to emulate it.

### 2.1 PRELIMINARIES

For each token position $t$ and sequence $x_{<t}$, let $\pi_{\text{base}}(\cdot \mid x_{<t})$ and $\pi_{\text{RL}}(\cdot \mid x_{<t})$ denote the conditional next-token distributions of the base and RL models, respectively. To quantify distributional differences, we use the Jensen–Shannon (JS) divergence, defined as

$$D_{\text{JS}}(\pi_{\text{base}}(\cdot \mid x_{<t}) \parallel \pi_{\text{RL}}(\cdot \mid x_{<t})) = \tfrac{1}{2}D_{\text{KL}}(\pi_{\text{base}}(\cdot \mid x_{<t}) \parallel M_t) + \tfrac{1}{2}D_{\text{KL}}(\pi_{\text{RL}}(\cdot \mid x_{<t}) \parallel M_t),$$

where $M_t = \tfrac{1}{2}\big(\pi_{\text{base}}(\cdot \mid x_{<t}) + \pi_{\text{RL}}(\cdot \mid x_{<t})\big)$.

One could use any notion of divergence or distance between probability measures, but we use JS divergence over something like KL divergence because: (i) it is symmetric, avoiding directional considerations; (ii) it is bounded in $[0, \log 2]$, preventing extreme values from dominating aggregate statistics; and (iii) it remains well-defined even when the measures lack absolute continuity with respect to each other. The latter is particularly important in practice, as memory constraints often limit the retrieval of the full distribution across the entire vocabulary, and also when comparing top-$p$ truncated distributions, for which KL divergence may be undefined.

Unless otherwise stated, divergences are computed on top-$p$ truncated distributions using the same sampling configuration employed during generation, while entropy and probabilities are from the

full estimated distributions. Robustness checks across different top-$p$ values and against estimates of the full distributions are provided in Appendix A.4 (Figures 26 and 28), as well as results on additional models/datasets.

## 2.2 DISTRIBUTION SHIFTS ARE HIGHLY TARGETED AND SPARSE

A natural starting question is: *how broadly are distributional shifts distributed across token positions?* To answer this, we examine the token-level JS divergence between the base and RL-finetuned models. Figure 1 percentile curves for DAPO and SimpleRL on their generated responses for AIME 2024.

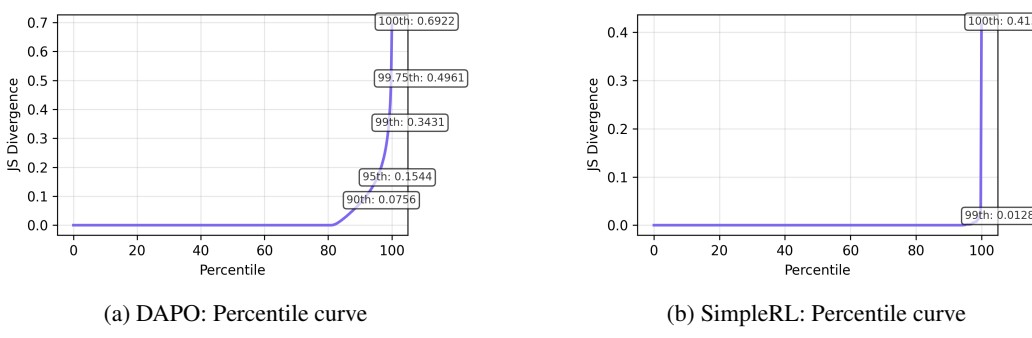

(a) DAPO: Percentile curve

(b) SimpleRL: Percentile curve

Figure 1: JS divergence distributions for Qwen2.5 32B DAPO and SimpleRL on AIME 2024.

The results reveal that **RLVR refinement is *highly sparse* at the token level.** Under DAPO, more than 83% of token positions exhibit near-zero divergence, while this proportion exceeds 98% under SimpleRL, indicating that only a small subset of token positions undergo substantial distributional change. Comparing the two, DAPO exhibits a broader divergence distribution and a more gradual percentile curve, consistent with its clip-higher mechanism and lack of KL regularization, permitting broader exploratory updates. In contrast, SimpleRL imposes stricter constraints, resulting in more tightly concentrated changes. Notably, even in the absence of KL regularization, the DAPO policy maintains near-zero divergence at most token distributions. For a more controlled comparison for models fine-tuned on the same dataset, Appendix A.4.2 presents the results for Qwen2.5-Math-7B trained with DAPO, comparing upper clip settings of 0.28 and 0.2. We see that, analogous to the results of the 32B models, the more restrictive 0.2 upper clip setting results in sparser distributional shifts, as shown by the percentiles corresponding to near-zero divergence (Figure 30).

## 2.3 POSITIONAL CONCENTRATION

Beyond how broadly changes are distributed across token positions, we next ask: *where within a generated sequence do distributional shifts tend to occur?* Figure 2 plots the mean and median JS divergence as a function of normalized token position (token index divided by response length), with percentile bands, for DAPO and SimpleRL on AIME 2024.

Both models exhibit a clear positional structure: average divergence is consistently highest near the beginning of the response, decreases through the middle, and increases modestly again toward the end. The early concentration aligns with the modification of initial high-level branching decisions, while the late increase aligns with adjustments to answer formatting and termination behavior. However, high divergence can occur throughout the response, as reflected in the wide percentile spread.

## 2.4 DIVERGENCE–ENTROPY RELATIONSHIP

To further understand the general structure underlying these sparse distributional shifts, we ask: *How are such shifts related to the model's token-level entropy?* We thus examine the relationship between distributional divergence and predictive entropy on the token level. For each token position $t$, we compute the token-level entropy and analyze how entropy relates to the distributional shifts from the base to RL model. Prior work suggests that RLVR updates may primarily affect high-entropy (uncertain) predictions while leaving low-entropy (confident) predictions largely unchanged

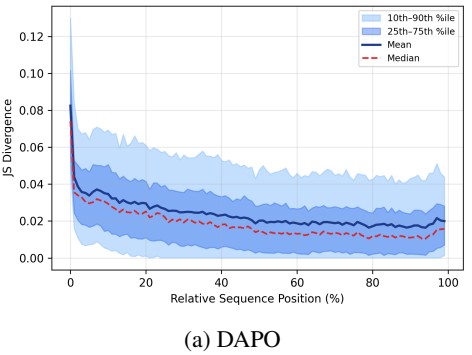 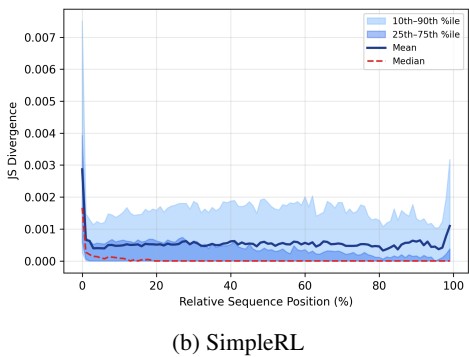

(a) DAPO                                    (b) SimpleRL

Figure 2: Mean and median JS divergence by normalized token position, with percentile bands. Both methods concentrate updates at the start and, to a lesser degree, at the end of responses.

(Wang et al., 2025). We explore this perspective by comparing entropy distributions across low- and high-divergence token positions. Specifically, token positions are grouped into low- and high-divergence bins, and we compare the entropy distributions of both the base and RL models within each bin. Figure 3 shows these results for DAPO, with corresponding SimpleRL results provided in Appendix A.4 (Figure 16).

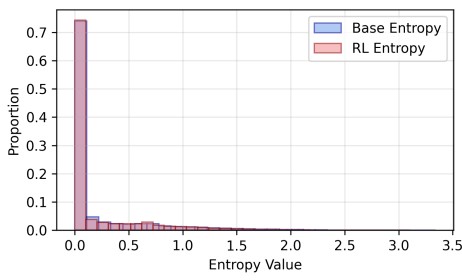 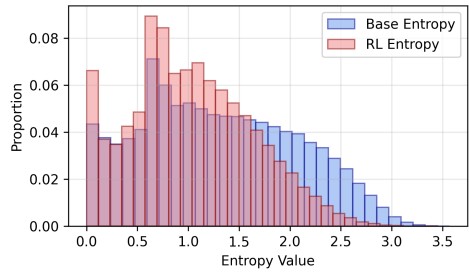

(a) Low JS divergence distributions ($< 0.1$).     (b) High JS divergence distributions ($> 0.1$).

Figure 3: Entropy distributions for low and high divergence distributions for **DAPO**. Low-divergence tokens are generally low-entropy, while high-divergence tokens span both high- and low-entropy regions, indicating that DAPO can modify even initially confident predictions.

The results show that low-divergence token distributions are largely low-entropy, indicating that distributions that are preserved are mostly initially low-entropy. High-divergence contexts, however, can span a broad entropy range. In particular, DAPO **modifies both initially high- and low-entropy predictions, demonstrating its ability to override even confident base-model outputs.** By contrast, SimpleRL concentrates divergence more strongly in higher base entropy regions, reflecting a more conservative update regime. Isolating the effect of clip-higher, Figure 35 illustrates this contrast more clearly. At high-divergence positions, the higher $0.28$ upper clip produces a greater proportion of distributions with low base entropy, whereas the $0.2$ clip concentrates its high-divergence distributions in the higher base entropy regime. Notably, the resulting RL entropy is higher under clip-higher, while for the $0.2$ clip it is concentrated at lower values, consistent with the entropy collapse observed under standard clipping (Yu et al., 2025) and the steadily increasing entropy induced by clip-higher.

## 2.5 SEMANTIC IDENTITY OF DIVERGENT TOKENS

Given the sparsity and general structure of these shifts, a natural next question is: *Which types of tokens are actually being targeted by RL fine-tuning?*

To investigate this, we examine which types of tokens tend to exhibit high versus low distributional divergence. Figure 4 visualizes representative examples using word clouds, where the size

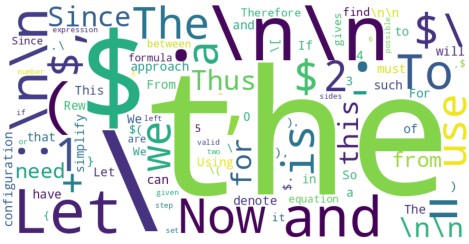
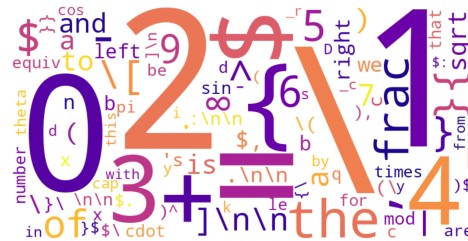

(a) Tokens with high JS divergence (JS > 0.1).   (b) Tokens with low JS divergence (JS < 0.01).

Figure 4: Word clouds of high and low divergence tokens under DAPO.

of each token is proportional to its frequency. Upon an initial examination, tokens appearing in high-divergence distributions include common function words, reasoning-related terms, and certain equation fragments, whereas those in low-divergence distributions are dominated by numerals, operators, and structural components of mathematical expressions.

**However, token identity alone does not determine divergence behavior**. Figure 19 shows the full JS divergence distributions for the tokens sampled most frequently from high- and low-divergence distributions, revealing substantial context dependence. For example, the word "the" appears among the most frequent high-divergence tokens, yet its full divergence distribution is overwhelmingly concentrated in the lower regime. This suggests that token identity alone is insufficient to characterize divergence, and that a contextual perspective is essential, rather than solely by token semantics.

## 2.6    COMPARISON WITH SUPERVISED FINE-TUNING (SFT)

While the above analyses reveal that RLVR induces sparse distributional shifts, it remains unclear whether this behavior is unique to RL fine-tuning. This raises the question: *Is such sparsity a distinctive property of RLVR, or a more general feature of fine-tuning methods?* A natural point of comparison is supervised fine-tuning (SFT), which optimizes models to imitate target tokens rather than optimizing verifiable rewards on self-generated trajectories. Appendix A.3 presents a controlled comparison between supervised fine-tuning (SFT) and RLVR (DAPO) on Qwen2.5-32B. Under the same JS divergence measurements (Section 2.2), SFT exhibits a substantially larger high-divergence set and a broader divergence distribution than RLVR (Figure 8). This demonstrates that the **sparsity of distributional shifts observed under RLVR is not a generic consequence of fine-tuning.**

## 3    CROSS-SAMPLING: FUNCTIONAL IMPORTANCE OF DIVERGENT DISTRIBUTIONS

In the previous section, we showed that, when evaluated along RL-generated trajectories, only a small fraction of token distributions exhibit substantial shifts between the base and RL models. This observation motivates a fundamental question: *Are these divergent token distributions directly responsible for the performance gains induced by RLVR? More generally, to what extent are the base and RL policies functionally different on their entire sequence distributions?*

To answer these questions, we conduct controlled *cross-sampling* experiments that selectively swap token choices between the base model $\pi_{\text{base}}$ and the RL-trained model $\pi_{\text{RL}}$. We consider two complementary interventions: (i) *forward cross-sampling*, which injects RL-sampled tokens into base-model generations, and (ii) *reverse cross-sampling*, which replaces RL-sampled tokens with base-model tokens during RL generation. Together, these interventions probe both the sufficiency and the necessity of RL-induced token-level changes. The general implementation procedure is summarized in Algorithm 1 of Appendix A.5.

### 3.1    CROSS-SAMPLING FRAMEWORK

Let $(X_t)_{t \geq 1}$ denote the sequence of random variables generated during decoding, and define the stopping time $\tau := \inf\{t \geq 1 : X_t = \text{EOS}\} \wedge T_{\max}$, where EOS is the end-of-sequence token

and $T_{\max}$ is the maximum number of tokens to generate. The generated response is then the finite sequence $X_{1:\tau}$. Let $\pi_{\mathrm{prim}}$ denote the *primary policy*, which governs generation by default, and let $\pi_{\mathrm{int}}$ denote the *intervention policy*, which is used only at selected positions. These policies induce sequence-level distributions $P_{\mathrm{prim}}$ and $P_{\mathrm{int}}$ over finite sequences. To model cross-sampling, we introduce a *switching rule* $\mathcal{S} : \mathcal{V}^{<\mathbb{N}} \to \{0,1\}$, where $\mathcal{V}^{<\mathbb{N}}$ is the set of finite sequences over the vocabulary $\mathcal{V}$, which determines, at each generation step, whether the next token is sampled from the $\pi_{\mathrm{int}}$ ($S_t = 1$) or $\pi_{\mathrm{prim}}$ ($S_t = 0$). Given a partial sequence $X_{<t}$, we define the switching variable $S_t := \mathcal{S}(X_{<t}) \in \{0,1\}$, and the resulting mixed policy governing the law of the next token:

$$X_t \sim \pi_{\mathrm{mix}}^{(\mathrm{prim},\mathrm{int})}(\cdot \mid X_{<t}) = (1 - S_t)\,\pi_{\mathrm{prim}}(\cdot \mid X_{<t}) + S_t\,\pi_{\mathrm{int}}(\cdot \mid X_{<t}).$$

The corresponding sequence-level distribution is then denoted by $P_{\mathrm{mix}}^{(\mathrm{prim},\mathrm{int})}$.

In our experiments, to align with the analysis in Section 2, the switching rule $\mathcal{S}$ is defined as $\mathcal{S}(X_{<t}) = \mathbb{1}\{D_{\mathrm{JS}}(\pi_{\mathrm{prim}}(\cdot \mid X_{<t}) \parallel \pi_{\mathrm{int}}(\cdot \mid X_{<t})) > \varepsilon\}$, given a fixed threshold $\varepsilon \geq 0$, so that cross-sampling intervenes only at high-divergence positions. In Appendix A.6, we provide simple bounds on divergences between the sequential distributions $P_{\mathrm{mix}}$ and $P_{\mathrm{int}}$ under different cross-sampling settings.

**Forward Cross-Sampling.**   In forward cross-sampling, the response is generated primarily under the base policy, which serves as the *primary policy*, i.e., $\pi_{\mathrm{prim}} = \pi_{\mathrm{base}}$. The *intervention policy* is the RL policy, $\pi_{\mathrm{int}} = \pi_{\mathrm{RL}}$. This procedure tests whether selectively injecting RL token choices into trajectories that are otherwise generated by the base model is sufficient to recover RL-level performance.

**Reverse Cross-Sampling.**   In reverse cross-sampling, the roles of the base and RL policies are reversed. The response is generated primarily under the RL policy, which serves as the *primary policy*, i.e., $\pi_{\mathrm{prim}} = \pi_{\mathrm{RL}}$, while the *intervention policy* is the base policy, $\pi_{\mathrm{int}} = \pi_{\mathrm{base}}$. This intervention selectively replaces RL-sampled tokens at high-divergence positions with base-model choices, allowing us to quantify how rapidly RL performance degrades when those decisions are removed.

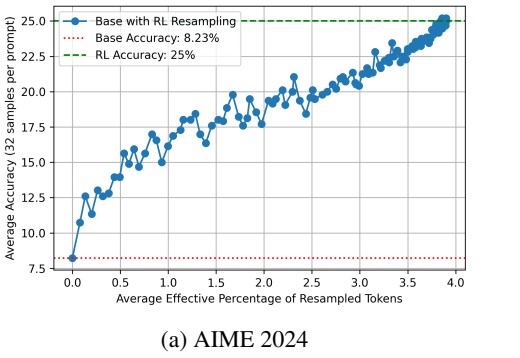 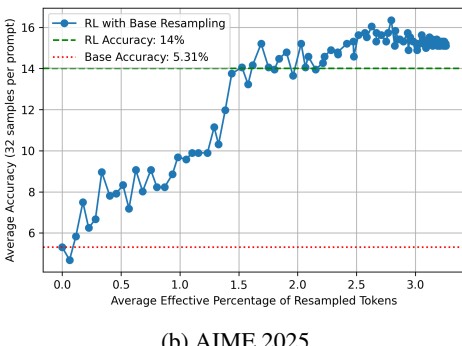

(a) AIME 2024                                              (b) AIME 2025

Figure 5: Forward cross-sampling results (Qwen2.5 32B SimpleRL): injecting RL tokens into base generations progressively recovers RL accuracy.

## 3.2   RESULTS AND FINDINGS

Figures 5 and 6 show the accuracy curves for forward and reverse cross-sampling on Qwen2.5-32B fine-tuned with SimpleRL, evaluated on AIME 2024. Each point along the curve corresponds to the Mean@32 accuracy obtained by generating responses under the mixed policy $P_{\mathrm{mix}}$ with a fixed number of interventions, after which generation is completed under the primary policy. This can be viewed as enforcing an intervention budget by setting $S_t = 0$ for all $t$ such that $\sum_{s=1}^{t-1} S_s \geq k$, where $k$ denotes the maximum number of cross-sampling interventions. This setup probes whether early, limited interventions are sufficient to induce downstream performance differences when generation is subsequently completed by the primary policy.

Table 1 summarizes the number and proportion of cross-sampled tokens required to approximately recover RL-level performance (forward) or collapse to base-level performance (reverse) for Qwen2.5 32B SimpleRL and DAPO on AIME 2024 and AIME 2025. Additional cross-sampling results and discussion, including experiments on additional model configurations and datasets, are provided in Appendix A.5.

> **Forward Cross-Sampling:** *A small fraction of RL tokens suffices to recover RL-level performance in the base model.*

**Forward Cross-Sampling:** From Figure 5 and Table 1, forward cross-sampling under the mixed policy $\pi_{\text{mix}}^{(\text{base,RL})}$ recovers RL-level accuracy with remarkably few interventions. Injecting fewer than $4\%$ RL-sampled tokens per sequence on average, corresponding to fewer than $40$ effective token substitutions per response, is sufficient to close the performance gap from the base model (approximately $8\%$) to the RL model (approximately $25\%$) on AIME 2024. On AIME 2025, the effect is even more pronounced: using only $1.53\%$ effective tokens, or $13$ average token substitutions per response, boosts accuracy from about $5\%$ to over $14\%$. Interestingly, this level of performance exceeds that of the RL policy $\pi_{\text{RL}}$ itself, indicating that the mixed policy $\pi_{\text{mix}}^{(\text{base,RL})}$ can outperform the standalone RL policy. This potentially occurs because the mixed policy is close but not identical to the RL policy (for $\varepsilon > 0$), which may sometimes avoid failures induced by the RL model. Recovering RL-level performance for DAPO requires a larger number of interventions, reflecting its substantially stronger fine-tuned performance. Importantly, even though the performance gains are substantially larger, **the number of critical token-level decisions remains small relative to sequence length.**

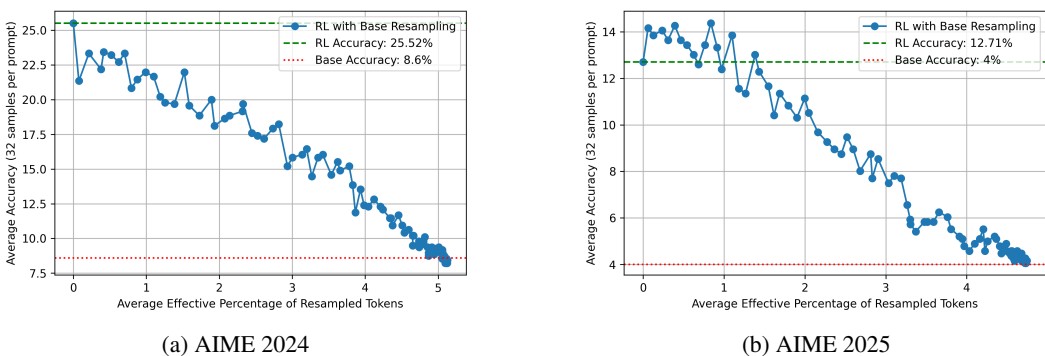

(a) AIME 2024                                (b) AIME 2025

Figure 6: Reverse cross-sampling results (Qwen2.5 32B SimpleRL): swapping RL tokens with base tokens in RL generations causes near-monotonic degradation toward base performance.

> **Reverse Cross-Sampling:** *Reverting a small portion of RL tokens collapses performance to base levels with the RL model.*

**Reverse Cross-Sampling:** Reverse cross-sampling shows that the RL policy is highly sensitive to a small number of its token-level decisions. From Figure 6 and Table 1, we observe that for Qwen2.5-32B with SimpleRL, generating under the mixed policy $\pi_{\text{mix}}^{(\text{RL,base})}$, only a small fraction of base-sampled tokens is enough to rapidly degrade performance. On AIME 2024, reverting approximately $5\%$ of high-divergence distributions, corresponding to less than $30$ effective base-sampled tokens per response, is enough to collapse accuracy from RL levels (around $25\%$) back to base-level performance (around $8\%$). This phenomenon also holds on AIME 2025: roughly $4.7\%$ of effective base-sampled tokens (around $30$ effective tokens per response) reduces accuracy from approximately $12.7\%$ to below $4\%$, which is below base-level performance. For DAPO, a larger number of substitutions is required to erase its gains, consistent with its substantially stronger performance. Importantly, even in these cases, **the required reversions constitute a small fraction**

**of the total generated tokens**, reinforcing that RL-level performance, across both SimpleRL and DAPO, depends critically on a sparse set of token-level decisions.

Notably, **the substituted base tokens are mostly locally plausible and semantically reasonable (Figure 43), yet they nonetheless progressively derail the reasoning process.** Even when two token choices are locally equivalent or interchangeable to a human reader, they can induce different downstream conditional distributions and lead to diverging reasoning responses, revealing trajectory sensitivity of the model.

Table 1: Summary of cross-sampled tokens required to reach approximate RL-level (forward) or base-level performance (reverse) for Qwen2.5-32B on AIME24 and AIME25 with a token generation budget of 8000. Effective token counts/percentages exclude identity swaps during cross-sampling. Token percentages are computed at the sequence level.

| Dataset | Method | Eff. % Tokens | Eff. # Tokens | Initial Acc. (%) | Final Acc. (%) |
|---|---|---|---|---|---|
| AIME24 | SimpleRL | 3.86% | 38 | 8.23 | $> 25$ |
| | SimpleRL Reverse | 5% | 29 | 25.52 | $< 8.3$ |
| | DAPO | 7.8% | 280 | 8.23 | $> 44$ |
| | DAPO Reverse | 10.1% | 173 | 44.8 | $< 8.5$ |
| AIME25 | SimpleRL | 1.53% | 13 | 5.3 | $> 14$ |
| | SimpleRL Reverse | 4.73% | 31 | 12.71 | $< 4$ |
| | DAPO | 6.47% | 230 | 5 | $> 33$ |
| | DAPO Reverse | 9.89% | 181 | 32 | $< 4.5$ |

# 4 FINE-GRAINED MECHANICS OF DISTRIBUTION SHIFTS

Having established the general aspects of RLVR-induced distribution shifts, namely their sparsity, positional concentration, relationship to entropy, and importance to downstream task performance, a natural next question is: *At token positions where substantial changes occur, how novel are these updates? Do they introduce (effectively) new candidate tokens, or primarily redistribute probability mass among existing ones?*

This fine-grained analysis reveals that, even at positions with substantial divergence, current RLVR methods mostly do not fundamentally change the candidate space of predictions. Instead, they primarily reorder and selectively amplify tokens that are already plausible under the base model, with limited promotion of low-probability tokens. We study this behavior through multiple lenses: (i) overlap in top-$k$ candidate sets and token rank reordering, (ii) low-probability token behavior, and (iii) the evolution over the course of training. We further validate these findings across additional datasets, model architectures, and RLVR hyperparameter settings Appendix A.4, demonstrating their robustness beyond the primary experimental configuration.

## 4.1 TOP-$k$ OVERLAP AND RANK REORDERING

We first investigate whether RLVR changes *which* tokens are considered plausible, or mainly changes *how* they are prioritized. Concretely, we examine (1) the overlap between the base and RL models' top-$k$ candidate sets, and (2) how the relative ranking of shared candidates shifts.

Figure 7 reports the fraction of shared tokens between the top-$k$ sets of the base and RL fine-tuned models, restricted to high-divergence token distributions. Despite only considering high-divergence positions, overlap remains high once $k \geq 2$. SimpleRL exhibits over 80% average overlap (often exceeding 85%), while DAPO shows slightly lower but still substantial overlap. Both methods display a sharp increase in overlap from $k = 1$ to $k = 2$, suggesting that while the top-1 token often changes, the replacement was typically already among the base model's top-3.

This observation is further clarified in Figure 17, which shows where the RL model's top-3 tokens appear in the base model's ranking, among high divergence positions. Around 30% of RL top-1 tokens are already ranked first under the base model, and over 80% (DAPO) and 90% (SimpleRL) fall within the base top-3. RL top-2 tokens typically lie within the base top-3–4, with SimpleRL exhibiting consistently stronger alignment.

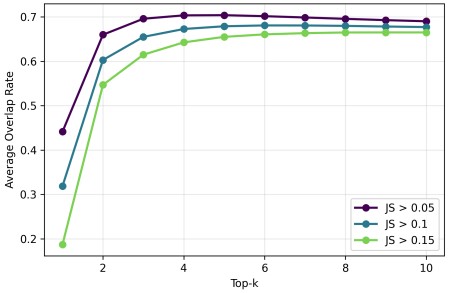 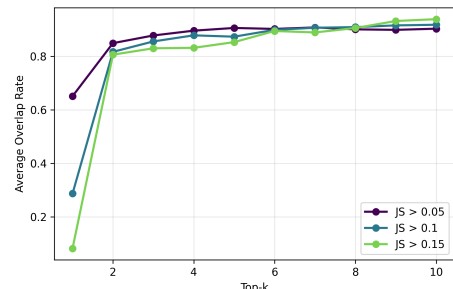

(a) DAPO: Top-$k$ overlap across thresholds.  (b) SimpleRL: Top-$k$ overlap across thresholds.

Figure 7: Top-$k$ token overlap between base and RL models at divergent positions ($JS > 0.1$). Computed as the size of the intersection divided by $k$. High overlap for $k \geq 2$ shows that distributional shifts occur mostly within shared candidate sets.

## 4.2 LOW-PROBABILITY BEHAVIOR: DOES RL INVENT OR SELECT?

We next examine whether RLVR promotes tokens that were highly unlikely under the base model, or instead amplifies alternatives that were already plausible but underweighted. For each high-divergence position, we take the RL model's top-1 token and record its probability under the base distribution. We then compute the fraction of such tokens whose base probability falls below a given threshold among high-divergence positions. Figure 18 shows that under DAPO, only about 5% of divergent top-1 tokens have base probability below $0.01$, while under SimpleRL this fraction is nearly zero (Figure 20). Thus, even for DAPO, which encourages broader exploration through its clip-higher mechanism and lack of KL regularization, RLVR rarely elevates tokens that were unlikely in the base model. Comparing DAPO variants (Figure 34), we observe that the clip-higher mechanism substantially increases the fraction of RL top-1 tokens (among high-divergence positions) whose base-model probability was initially very small, relative to the variant without clip-higher. This aligns with earlier observations and supports the interpretation that clip-higher enables greater exploration, allowing tokens that were low probability in the base model distribution to be promoted more frequently. Importantly, although such low-probability promotions remain rare overall, they may still be consequential and important for improved reasoning performance.

## 4.3 EVOLUTION ACROSS TRAINING

Finally, we analyze how the distributional shifts develop over training. Using intermediate checkpoints from DAPO training on Qwen2.5-Math-7B, we track token-level distributions while conditioning on the final model's outputs, allowing us to follow the evolution of divergence for a fixed set of token sequences. Figure 29 shows that JS divergence increases monotonically throughout training, with higher percentiles (e.g., 95th and 99th) growing faster than lower ones. This widening gap indicates that distributional change becomes increasingly concentrated in a small subset of tokens, while the majority remain relatively stable. Consistent with this perspective, the Jaccard overlap between each checkpoint's divergent-token set and the final set increases gradually before rising sharply near the end of training (Figure 29b).

## 5 RELATED WORK

**RL fine-tuning in LLMs.** Reinforcement learning has become an important component of LLM fine-tuning, stemming from reinforcement learning with human feedback (RLHF) used to align LLM behavior to human preferences (Christiano et al., 2023; Ouyang et al., 2022). Recently, RLVR has emerged as a central paradigm for improving *reasoning* by optimizing with verifiable reward signals of generated responses (Lambert et al., 2024). A number of RLVR methods build on policy-gradient variants, including Group Relative Policy Optimization (GRPO) (Shao et al., 2024), as well as extensions such as Dr.GRPO (Liu et al., 2025b), DAPO (Yu et al., 2025), and sequence-level variants such as Group Sequence Policy Optimization (GSPO) (Zheng et al., 2025). Beyond these core methods, several works propose complementary improvements to target impactful updates and

stabilize training, including entropy-based perspectives (Wang et al., 2025; Cheng et al., 2025), clipping/KL regularization strategies (Cui et al., 2025), reweighting based on token probability, perplexity, or position (Yang et al., 2025; Deng et al., 2025), as well as analyses of different reward designs (Shao et al., 2025). Additional works study the reasoning boundaries of RLVR and how to expand it (Yue et al., 2025; Wen et al., 2025; Liu et al., 2025a).

**Understanding RLVR and its differences with SFT.** A growing body of work argues that RL fine-tuning often acts as a *scalpel rather than a hammer*, amplifying existing capabilities through localized changes. This is supported through the perspective of evaluations on different domains/-tasks, catastrophic forgetting, parameter changes, and overall KL divergence (Rajani et al., 2025; Chu et al., 2025; Shenfeld et al., 2025; Huan et al., 2025). Recent work also analyze locality from the *parameter* perspective: for example, Mukherjee et al. (2025) report that RL fine-tuning concentrates effective updates into small subnetworks, while Zhu et al. (2025) provide theoretical insight into RLVR learning dynamics and the structure of these parameter-sparse updates. Our paper complements these perspectives by focusing on quantifying the changes induced by RLVR at the level of *token distributions*. We show that RL not only induces smaller aggregate divergence than SFT, but that its changes are substantially sparser at the token level.

**Token-Level analyses of RLVR.** Several works seek to understand RLVR through a token-level lens. Wang et al. (2025) attribute a substantial portion of RL gains to high-entropy minority tokens, while Cheng et al. (2025) connect such tokens to exploratory reasoning steps; related work also highlights entropy collapse risks and token-level regularization mechanisms (Cui et al., 2025). Other studies emphasize the disproportionate role of specific tokens or sampling decisions (Vassoyan et al., 2025; Lin et al., 2025; Karan & Du, 2025), and Huan et al. (2025) analyzes RL-induced changes using token-level KL divergence and token rank shifts. Recent work by Chen et al. (2026) shows that RL training modifies a sparse subset of tokens when viewed through rank-shift statistics, and develops a theoretical analysis based on reasoning patterns. Our contributions are complementary but distinct: we conduct a systematic empirical study of RLVR-induced token-level changes through the lens of quantities such as divergence, entropy, and probability mass redistribution, and connect these shifts to sequence-level reasoning via cross-sampling interventions that establish their impact on reasoning performance.

## 6    CONCLUSION

Our study reveals that reinforcement learning with verifiable rewards (RLVR) reshapes LLMs in a manner that is sparse, targeted, and structured rather than uniformly diffused across tokens. By analyzing token-level distributional shifts, we show that only a small subset of tokens undergo meaningful divergence, and that these divergences carry disproportionate functional importance: cross-sampling interventions confirm that performance gains hinge on precisely these positions. Moreover, our fine-grained analyses suggest that, even at high-divergence positions, RLVR typically refines behavior by reallocating probability mass within an existing candidate set rather than introducing fundamentally new tokens. At the same time, the comparatively rare cases of substantial re-ranking and promotion of initially low-probability tokens may still be important for the observed improvements in reasoning. To complement these analyses, we explored divergence-weighted advantage, a simple modification that scales token-level advantages by per-token divergence. These results suggest that such weighting strategies can influence learning dynamics, though stabilizing performance may require model-specific choices and further investigation.

Together, these findings advance a token-level understanding of RL fine-tuning. They highlight that the essence of RLVR's success lies not in widespread distributional changes, but in selective refinements aligned with varying entropy levels. Taken together, our findings suggest that RLVR operates not as a global policy shift, but as a sparse intervention on a small set of high-impact decision points that steer generation trajectories largely accessible to the base model. Beyond clarifying the mechanics of existing methods, our work offers a perspective for designing future RL objectives and diagnostics that explicitly incorporate distributional structure, opening avenues for more effective, interpretable, and controllable LLM post-training.

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

# A APPENDIX

## A.1 EXPLORATORY INVESTIGATION: DIVERGENCE-WEIGHTED ADVANTAGES

Our earlier analyses reveal that RL refinements are *sparse and targeted*, with only a small subset of tokens exhibiting meaningful distributional change. Moreover, cross-sampling experiments demonstrate that these high-divergence tokens are functionally critical, with performance gains hinging on precisely these positions. This raises a natural question: *if only a small fraction of tokens drive improvements, can training be more effectively guided by modulating token-level learning signals according to these divergences?* To investigate this possibility, we conduct a preliminary exploration of *divergence-weighted advantages* as a diagnostic intervention, where advantages are reweighted by token-level distributional divergence. We explore two different approaches: *high-KL boost*, which concentrates updates towards token distributions that are already changing substantially, and *low-KL boost*, which focuses updates on distributions that have changed less, potentially encouraging updates in previously stable regions.

### A.1.1 GRPO-BASED METHODS

**GRPO in brief.** GRPO (Shao et al., 2024) samples $G$ responses $\{o_i\}_{i=1}^G$ from a policy $\pi_{\theta_{\text{old}}}(\cdot \mid q)$ for a prompt $q$ with ground-truth answer $a$, assigns sequence-level rewards $\{R_i\}_{i=1}^G$, and computes a group-normalized advantage for each sample. GRPO then applies a PPO-style (Schulman et al., 2017) clipped surrogate objective at the token level, typically with an explicit KL penalty to a reference model.

**DAPO.** DAPO (Yu et al., 2025) modifies GRPO with an asymmetric clip-higher mechanism, dynamic sampling of correct/incorrect completions, token-level averaging, and removal of the explicit KL penalty term. Its objective is given by

$$J_{\text{DAPO}}(\theta) = \mathbb{E}_{(q,a)\sim\mathcal{D},\ \{o_i\}_{i=1}^G\sim\pi_{\theta_{\text{old}}}(\cdot|q)}$$

$$\left[ \frac{1}{\sum_{i=1}^G |o_i|} \sum_{i=1}^G \sum_{t=1}^{|o_i|} \min\left( r_{i,t}(\theta)\,\hat{A}_{i,t},\ \text{clip}\big(r_{i,t}(\theta),\, 1 - \epsilon_{\text{low}},\, 1 + \epsilon_{\text{high}}\big)\,\hat{A}_{i,t} \right) \right] \quad (1)$$

$$\text{s.t.} \quad 0 < \left| \left\{ o_i \mid \text{is\_equivalent}(a, o_i) \right\} \right| < G,$$

with

$$r_{i,t}(\theta) = \frac{\pi_\theta(o_{i,t} \mid q, o_{i,<t})}{\pi_{\theta_{\text{old}}}(o_{i,t} \mid q, o_{i,<t})}, \quad \hat{A}_{i,t} = \frac{R_i - \text{mean}\big(\{R_j\}_{j=1}^G\big)}{\text{std}\big(\{R_j\}_{j=1}^G\big)}. \quad (2)$$

### A.1.2 DIVERGENCE-WEIGHTED ADVANTAGE

Standard RLVR objectives treat all tokens within a sequence uniformly in terms of their advantages (though the importance sampling ratios are defined on the token-level). Motivated by our observation that distributional shifts are sparse and concentrated, we investigate whether modulating token-level advantages according to divergence magnitude can help improve or control aspects of training. We explore modifications where advantages are rescaled depending on the per-token divergences.

**General formulation.** We define a divergence-weighted advantage:

$$\tilde{A}_t = w_t \cdot \hat{A}_t, \quad (3)$$

where $\hat{A}_t$ denotes the standard group-normalized advantage and $w_t$ is a per-token weight based on divergence. To ensure that the introduced divergence weight influences only the weighting, divergences are detached from the computation graph.

**Choice of divergence.** For ease of compatibility with standard frameworks, we employ KL divergence with respect to the old policy as our primary divergence quantity:

$$\text{KL}_t^{\text{old}} = D_{\text{KL}}(\pi_{\theta_{\text{old}}}(\cdot \mid x_{<t}) \,\|\, \pi_\theta(\cdot \mid x_{<t})), \quad (4)$$

where $\pi_{\theta_{\mathrm{old}}}$ denotes the policy from the previous update iteration, as in PPO/GRPO. This old-policy KL quantifies the magnitude of recent policy updates at each token position, serving as a proxy for the extent of local distributional change. For computational efficiency and compatibility with existing training frameworks such as verl (Sheng et al., 2024), we estimate these quantities using KL estimators computed over sampled tokens only, which may not capture the full distributional structure. In particular, we use the $k_3$ estimator (Schulman, 2020) defined by

$$\mathrm{KL}_{\mathrm{est}}(\pi_{\theta_{\mathrm{old}}} \parallel \pi_\theta) \approx k_3 \left( \frac{\pi_\theta(\cdot \mid x_{<t})}{\pi_{\theta_{\mathrm{old}}}(\cdot \mid x_{<t})} \right) = \frac{\pi_\theta(\cdot \mid x_{<t})}{\pi_{\theta_{\mathrm{old}}}(\cdot \mid x_{<t})} - \log \frac{\pi_\theta(\cdot \mid x_{<t})}{\pi_{\theta_{\mathrm{old}}}(\cdot \mid x_{<t})} - 1.$$

Alternative divergences signals, including the divergence to the base model (capturing global shifts), are discussed in Appendix A.7.

**Weighting scheme.** We adopt a simple sigmoid weighting scheme (to ensure bounded weights), which transforms divergence into weights through:

$$w_t = 1 + s \left( \sigma(\alpha \cdot \mathrm{KL}_t) - \tfrac{1}{2} \right), \quad \sigma(x) = \tfrac{1}{1+e^{-x}}. \tag{5}$$

The parameter $\alpha$ controls the direction and magnitude of emphasis: $\alpha > 0$ amplifies high-divergence tokens, whereas $\alpha < 0$ emphasizes low-divergence ones. The sigmoid function provides a smooth, bounded nonlinear transformation that enables selective focus on either high- or low-divergence regions depending on the sign of $\alpha$. This formulation allows us to investigate whether concentrating the learning signal on regions that have already changed or those that remain unchanged yields more effective training dynamics. Alternative weighting schemes, including linear relative weighting, are dicussed in Appendix A.7.

**Evaluation.** We train with divergence-weighted advantages using the DAPO training recipe and data on Qwen2.5-Math-7B, evaluating on AIME 2024, AIME 2025, and AMC. Results are presented in Table 2. Detailed training hyperparameters and implementation details are documented in Appendix A.2.3.

Table 2: Accuracy (%) under divergence-weighted configurations on Qwen2.5-Math-7B. Results shown for KL divergence with $\pi_{\theta_{\mathrm{old}}}$ and sigmoid weighting scheme across AIME 2024, AIME 2025, and AMC datasets. The results displayed are the Mean@32 scores (or the pass@1 scores computed using 32 samples). Results are each averaged over 3 runs.

| Configuration | AIME24 | AIME25 | AMC | Overall Avg |
|---|---|---|---|---|
| Baseline DAPO | 33.61 | 18.75 | 75.08 | $42.48 \pm 1.35$ |
| Low-KL boost | 35.90 | 19.90 | 78.97 | $44.92 \pm 0.05$ |
| High-KL boost | 36.74 | 20.00 | 78.40 | $45.05 \pm 0.79$ |

These results demonstrate that weighting token-level updates by divergence can amplify performance gains, providing empirical support for the hypothesis that targeted tokens disproportionately drive improvements. Both low-KL and high-KL boost configurations yield improvements over the baseline, suggesting that different divergence weighting strategies can be effective. However, the optimal choice between these approaches, and indeed whether divergence weighting provides benefits at all, may depend on the specific models and training methods used. Effective divergence weighting across training configurations may require model-specific paradigms or adaptive scheduling mechanisms to stabilize learning dynamics. We present this approach as a complementary diagnostic tool that may inform future refinements of token-level training strategies. These include approaches that aggregate information from token-level divergences, more effectively promote the rare actions discussed in Section 4, and directly modify the clipping mechanism.

## A.2 EXPERIMENTAL DETAILS

### A.2.1 TOKEN ANALYSIS

**Models and Datasets.** Our primary analysis focuses on Qwen2.5-32B (Qwen et al., 2025) as the base model, with RLVR variants trained using DAPO (Yu et al., 2025) and GRPO, the latter paired

with the corresponding SimpleRL model (Zeng et al., 2025). For evaluation on AIME 2024 and AIME 2025, we sample 32 responses per problem for robustness. We further extend the analysis to additional models (Qwen2.5-Math-7B (Yang et al., 2024) with two variants corresponding to different upper clip settings, and Mistral-Small-24B (MistralAI, 2025)), datasets (AIME25, GPQA (Rein et al., 2023), and the models' respective fine-tuning datasets), and to comparisons with supervised fine-tuning (SFT). These extensions, reported in Appendix A.3 and Appendix A.4, confirm that our findings generalize across models, datasets, and training paradigms.

We run model inference using `vllm` (Kwon et al., 2023). On AIME, we apply nucleus sampling (Holtzman et al., 2020) with topp $= 0.7$ and temperature $= 1$. For divergence calculations on AIME, we use the top-$p$ truncated distribution to reflect the effective sampling distribution, to provide a more accurate estimate for our cross-sampling experiments. We also examine the distribution of JS divergence values without truncation (and on other topp values) to ensure the main results are not impacted by the truncation. For experiments on the fine-training data, we use topp $= 1$ to reflect the training sampling distribution.

### A.2.2 CROSS-SAMPLING

For cross-sampling experiments, we use the same inference setup as token analysis. Cross-sampling experiments selectively swap tokens between base and RL models at positions where JS divergence exceeds a threshold, allowing us to measure the functional importance of divergent token distributions.

We perform forward and reverse cross-sampling experiments on the following model-dataset combinations. For forward cross-sampling, we inject RL-sampled tokens into base generations at positions where JS divergence exceeds the specified threshold. For reverse cross-sampling, we replace RL tokens with base tokens at high-divergence positions on the RL generations. The divergence thresholds used for each configuration are as follows:

- **Qwen2.5-32B + SimpleRL:**
    - AIME 2024: Forward threshold $\varepsilon = 0.03$, Reverse threshold $\varepsilon = 0.05$
    - AIME 2025: Forward threshold $\varepsilon = 0.05$, Reverse threshold $\varepsilon = 0.05$
- **Qwen2.5-32B + DAPO:**
    - AIME 2024: Forward threshold $\varepsilon = 0.08$, Reverse threshold $\varepsilon = 0.06$
    - AIME 2025: Forward threshold $\varepsilon = 0.1$, Reverse threshold $\varepsilon = 0.08$
- **Mistral-Small-24B + SimpleRL:**
    - AIME 2024: Forward threshold $\varepsilon = 0.002$, Reverse threshold $\varepsilon = 0.02$

### A.2.3 ADDITIONAL TRAINING DETAILS

We implement RLVR training experiments using `verl` (Sheng et al., 2024) with the standard DAPO recipe (Yu et al., 2025).

**Qwen2.5-Math-7B DAPO Training.** We follow the public DAPO recipe, namely with clip ratios $\epsilon_{\text{low}} = 0.2$ and $\epsilon_{\text{high}} = 0.28$. However, for token analysis, we also train a variant with $\epsilon_{\text{high}} = 0.2$ for comparison. We optimize with learning rate $1 \times 10^{-6}$, a 10-step warmup using AdamW, and no explicit reference-KL penalty. Each RLVR step processes 512 prompts with 16 sampled responses per prompt; these are split into mini-batches of 32 prompts, yielding 16 gradient updates per RLVR step. Maximum generation length and the overlong-penalty threshold are set to 8k and 4k tokens.

**Supervised Fine-Tuning (SFT) Training.** For the SFT model based on on Qwen2.5 32B, we sampled 42k instances from the `AM-DeepSeek-R1-Distilled-1.4M` dataset. The model underwent full parameter fine-tuning for 5 epochs, employing DeepSpeed ZeRO-3 optimization.

**Divergence-weighted Training** For the divergence-weighted advantage experiments on Qwen2.5-Math-7B, under the **high-KL** setting we use $s = 0.3$ and set $\alpha$ to increase linearly from 0 to 50 starting at step 100. In the **low-KL** setting, we use $s = 0.3$ and set $\alpha$ to increase linearly from 0 to 50, which we linearly increase beginning at step 150.

For **Qwen2.5-7B**, in the high-KL relative setting we set $\alpha = 4$. In the configuration with an additional scheduler, we initialize $\alpha = 2$ and linearly increase it to 3 from step 80 onward.

### A.3 RLVR vs. Supervised Fine-Tuning: Contrasting Distributional Patterns

A natural question is whether the sparse, targeted distributional shifts we observe are specific to RLVR, or if they also characterize other fine-tuning approaches. To address this, we compare RLVR-trained models with models refined through supervised fine-tuning (SFT). We analyze Qwen2.5-32B trained with SFT alongside Qwen2.5-32B DAPO.

Figure 8 shows JS divergence distributions for both approaches. SFT produces a noticeably larger high-divergence set, whereas RLVR concentrates almost all token distributions below very small JS values. This directly reflects RLVR's extreme selectivity and the broader edits introduced by SFT. The top-$k$ overlap analysis (Figure 11) highlights that SFT consistently achieves lower overlap with the base model, indicating more aggressive re-ranking, while RLVR largely stays within the base model's existing candidate set. The rank reordering analysis (Figure 12) further shows that SFT promotes many more tokens far outside the base model's top-3, whereas RLVR mainly promotes candidates that were already high-ranked.

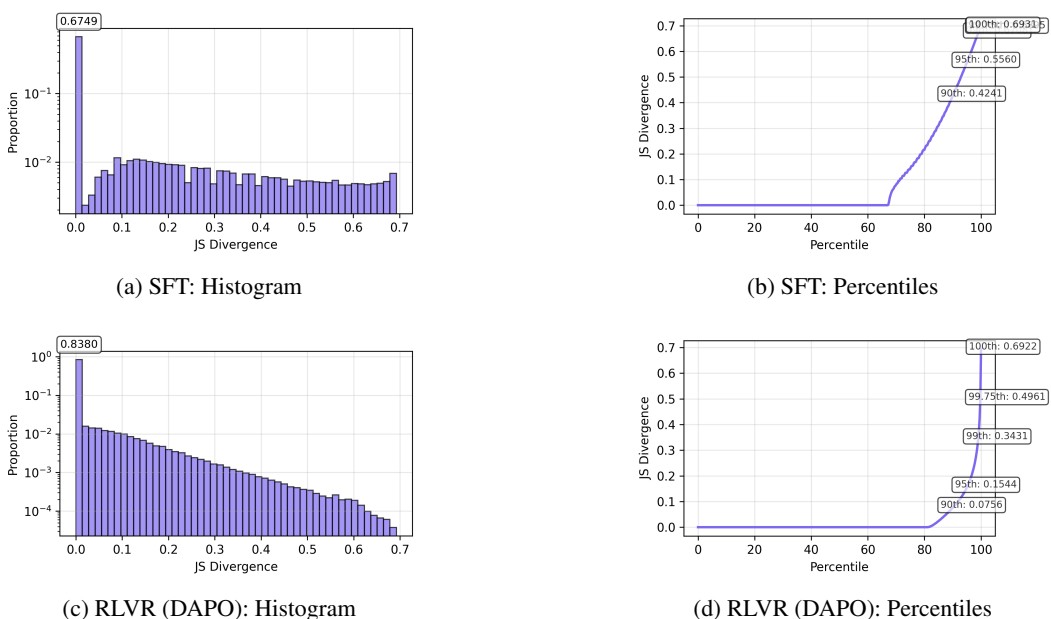

(a) SFT: Histogram

(b) SFT: Percentiles

(c) RLVR (DAPO): Histogram

(d) RLVR (DAPO): Percentiles

Figure 8: JS divergence distributions comparing SFT and RLVR on AIME 2024. RLVR exhibits even sparser distributional shifts than SFT, suggesting more targeted refinement.

Positional analysis further shows that SFT induces elevated divergence across the entire response, while still exhibiting increased divergence near the start of the sequence (Figure 10), mirroring, the early-position effects seen in RLVR. Finally, under the divergence–entropy analysis (Section 2.4), SFT's divergent tokens concentrate more strongly in regions of high base-model entropy (compared with DAPO). While this concentration may be partially influenced by SFT outputs appearing more uncertain when evaluated under the base model, the resulting fine-tuned entropy values are nevertheless substantially lower than those of the base model (Figure 13). These results are consistent with SFT's objective of directly learning target outputs, leading to globally broader and sharper distributional updates.

Taken together, the metrics highlight that SFT diverges from RLVR along several axes. The SFT model exhibits higher JS divergence overall as well as a larger mass of high-divergence tokens (Figure 8), and attains lower top-$k$ overlap with the base model (Figure 11) alongside larger rank shifts (Figure 12). These differences reinforce that RLVR acts as a targeted editor, while SFT drives broader, less selective reshaping of the distribution.

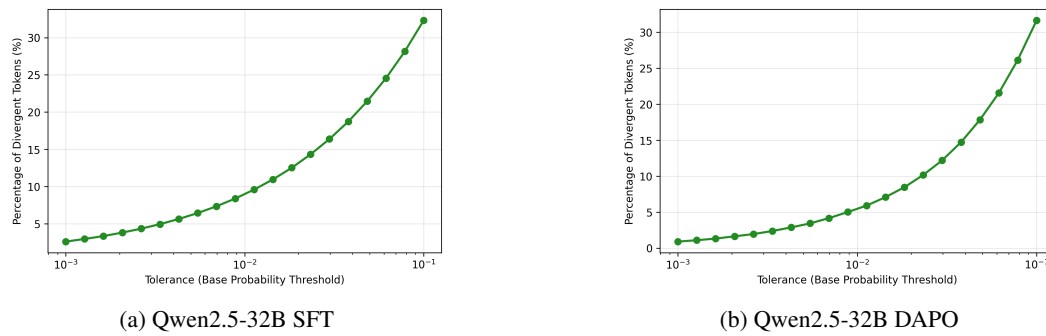

(a) Qwen2.5-32B SFT

(b) Qwen2.5-32B DAPO

Figure 9: Percentage of divergent tokens whose RL top-1 choice had base probability below a given threshold comparing SFT and RLVR-trained models on AIME 2024.

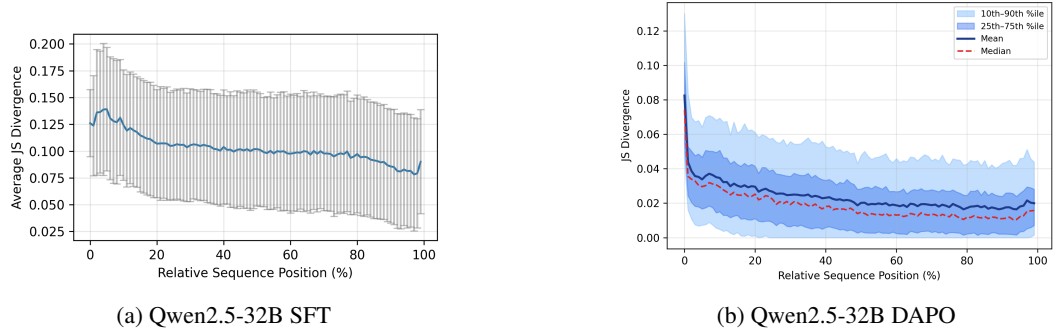

(a) Qwen2.5-32B SFT

(b) Qwen2.5-32B DAPO

Figure 10: Mean JS divergence by normalized token position comparing SFT and RLVR-trained models on AIME 2024.

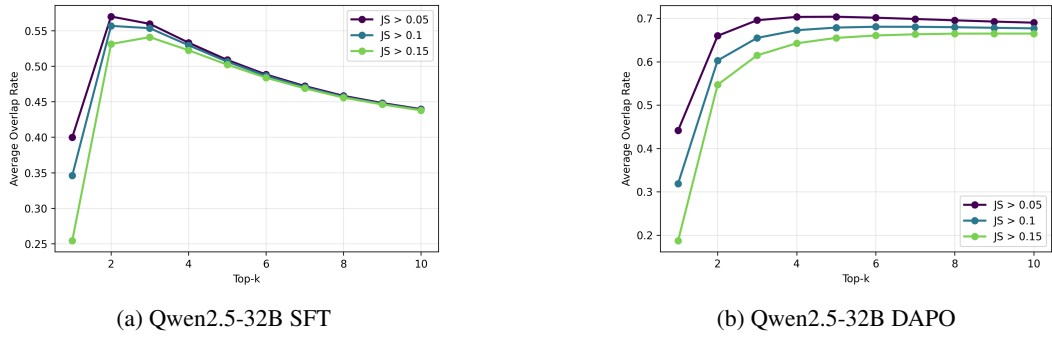

(a) Qwen2.5-32B SFT

(b) Qwen2.5-32B DAPO

Figure 11: Top-$k$ token overlap between base and refined models at divergent positions ($\mathrm{JS}_t > 0.1$) comparing SFT and RLVR-trained models on AIME 2024.

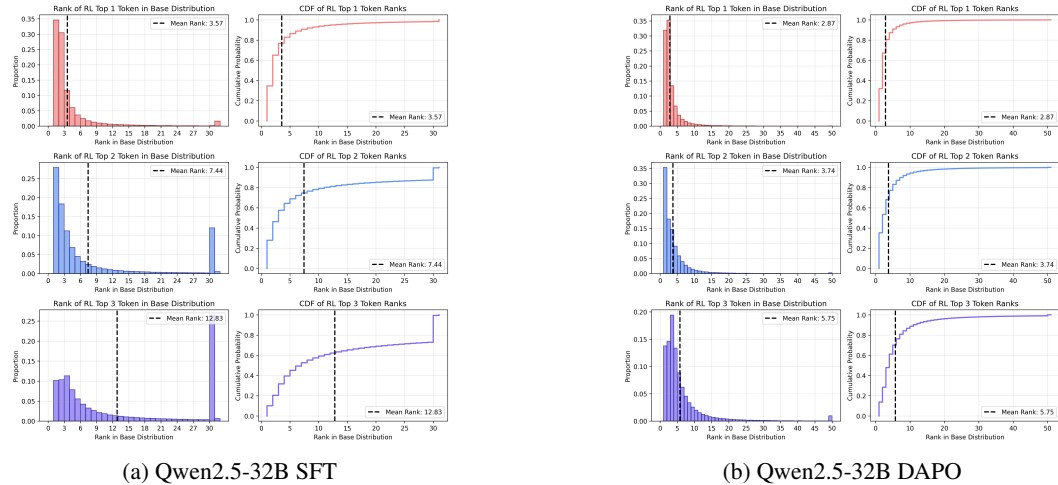

(a) Qwen2.5-32B SFT                    (b) Qwen2.5-32B DAPO

Figure 12: Distribution of base-model ranks for refined models' top-3 tokens at high-divergence positions ($JS > 0.1$) comparing SFT and RLVR-trained models on AIME 2024.

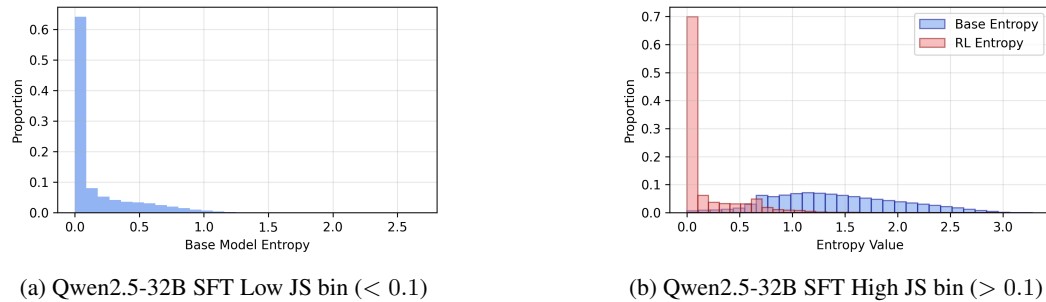

(a) Qwen2.5-32B SFT Low JS bin ($< 0.1$)        (b) Qwen2.5-32B SFT High JS bin ($> 0.1$)

Figure 13: Entropy distributions across divergence bins for Qwen2.5-32B SFT on AIME 2025.

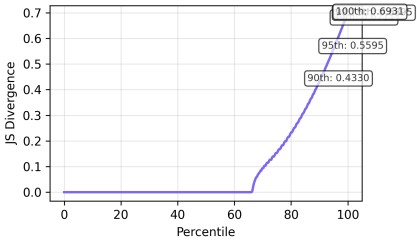

(a) Qwen2.5-32B SFT AIME 2025: Percentiles

Figure 14: JS divergence distributions for Qwen2.5-32B SFT on AIME 2025. Consistent patterns with AIME 2024 demonstrate robustness across datasets.

## A.4 Additional Token Distribution Analyses

This section provides supplementary and extended token distribution analyses. We first present supplementary figures for the main models (Qwen2.5-32B with DAPO and SimpleRL on AIME 2024), then extend the analysis to additional models and datasets to demonstrate the generalizability of our findings.

### A.4.1 Supplementary Figures for Main Models

We provide additional figures for Qwen2.5-32B with DAPO and SimpleRL that complement the analyses in the main text.

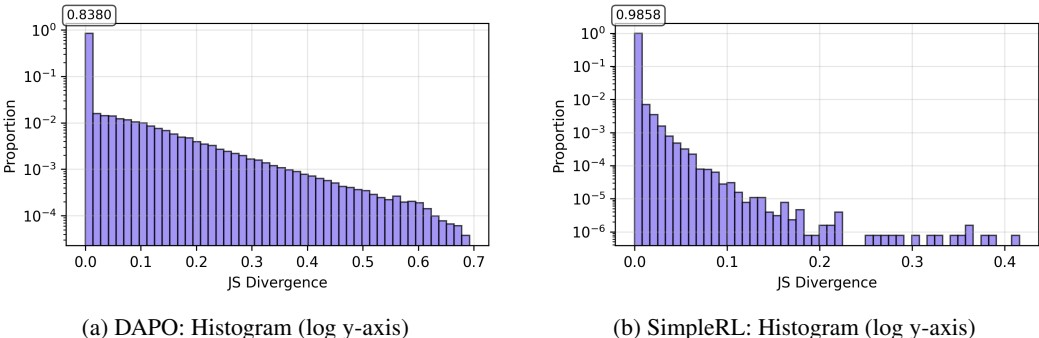

(a) DAPO: Histogram (log y-axis)

(b) SimpleRL: Histogram (log y-axis)

Figure 15: JS divergence distributions for Qwen2.5 32B DAPO and SimpleRL on AIME 2024.

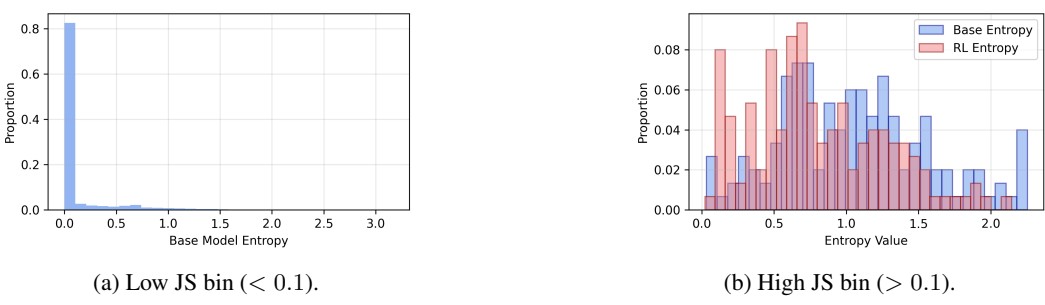

(a) Low JS bin ($< 0.1$).

(b) High JS bin ($> 0.1$).

Figure 16: Entropy distributions across divergence bins for **SimpleRL**. Low-divergence tokens are mostly low-entropy, while high-divergence tokens are concentrated in higher-entropy regions, reflecting a more conservative update strategy.

To supplement the positional analysis in the main text, we also examine localized averages of JS divergence near the start of the generation and near the final answer span.

**Results on GPQA-Diamond.** We extend our analysis to GPQA-Diamond to demonstrate the generalizability of our findings across different reasoning benchmarks. Figure 24 shows JS divergence percentile curves and positional concentration for Qwen2.5-32B with DAPO on GPQA-Diamond, revealing consistent sparsity patterns. Figure 25 shows entropy distributions across divergence bins.

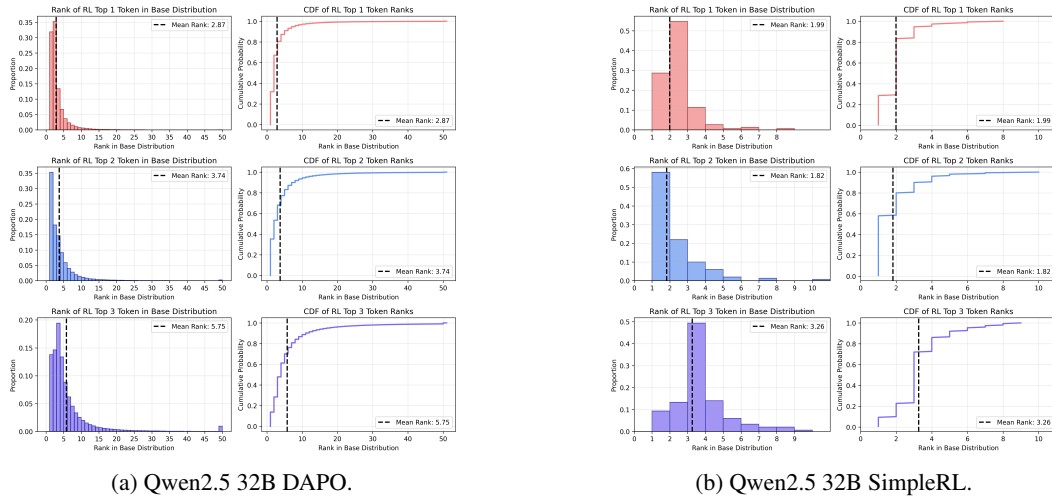

(a) Qwen2.5 32B DAPO.

(b) Qwen2.5 32B SimpleRL.

Figure 17: Distribution of base-model ranks for RL's top-3 tokens at high-divergence positions (JS > 0.1). Most RL-selected tokens were already highly ranked in the base model, especially under SimpleRL.

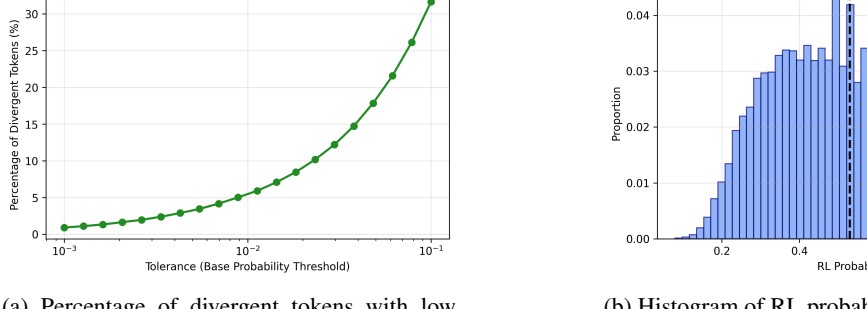

(a) Percentage of divergent tokens with low base probability.

(b) Histogram of RL probabilities for low base-probability tokens.

Figure 18: Analysis of tail behavior under DAPO for divergent token distributions (JS > 0.1). **(a)** shows the percentage of divergent tokens whose RL top-1 choice had base probability below a given threshold. **(b)** shows the distribution of RL probabilities for the subset with base probability < 0.01.

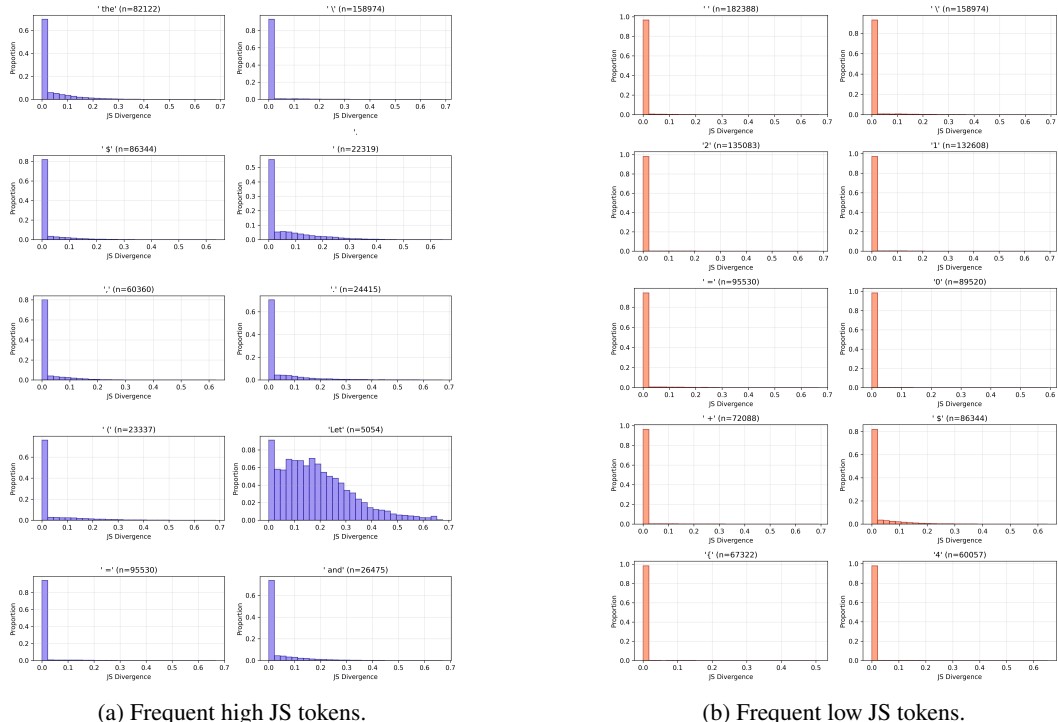

(a) Frequent high JS tokens.

(b) Frequent low JS tokens.

Figure 19: Histogram of divergences for frequent high JS tokens and frequent low JS tokens (Qwen2.5 32B with DAPO).

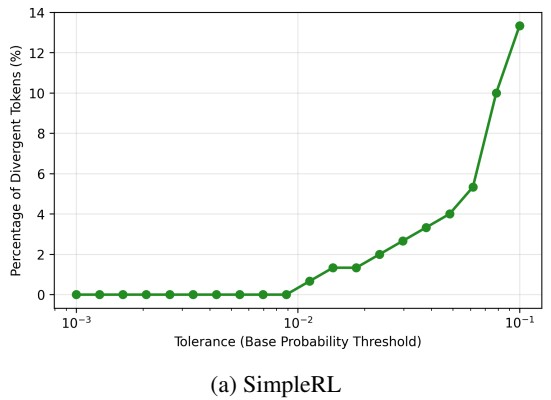

(a) SimpleRL

Figure 20: Percentage of divergent tokens whose RL top-1 choice had base probability below a given threshold.

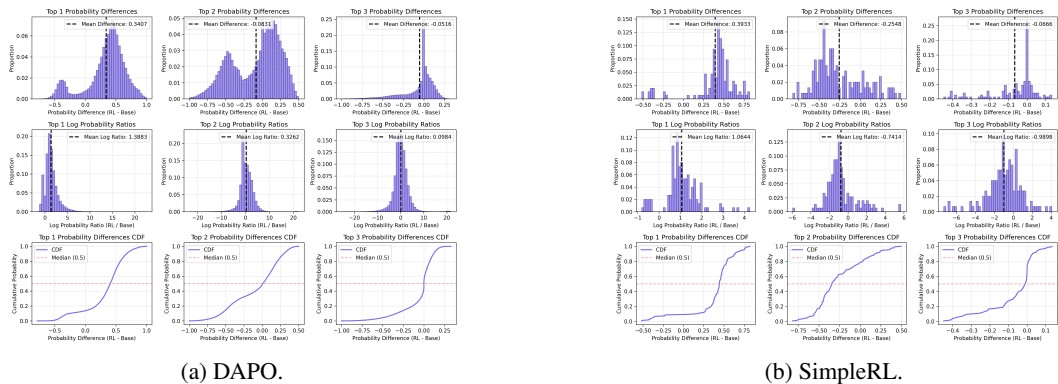

(a) DAPO.

(b) SimpleRL.

Figure 21: Probability differences and ratios for top-3 tokens under DAPO and SimpleRL among divergent distributions (JS > 0.1).

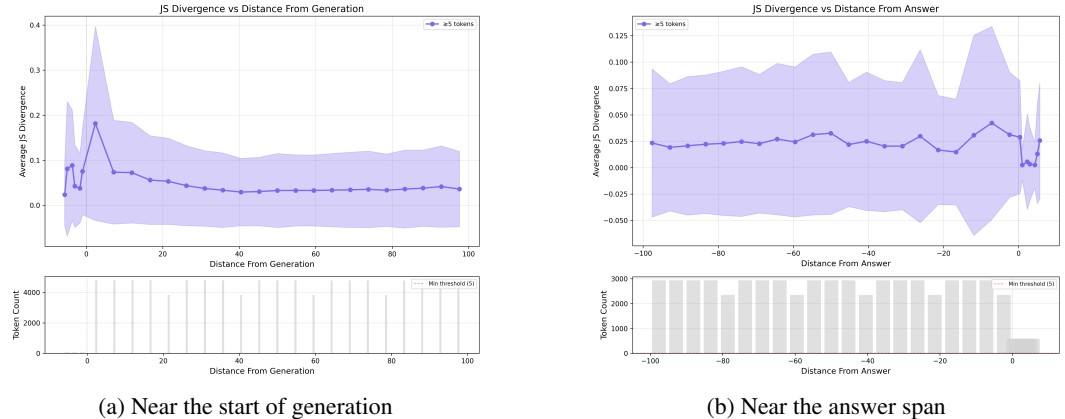

(a) Near the start of generation

(b) Near the answer span

Figure 22: Local averages of JS divergence as a function of distance from key regions (prompt beginning and answer) for Qwen2.5-32B models on AIME 2024. Average divergence peaks occur in the same early and late windows highlighted by the positional analysis.

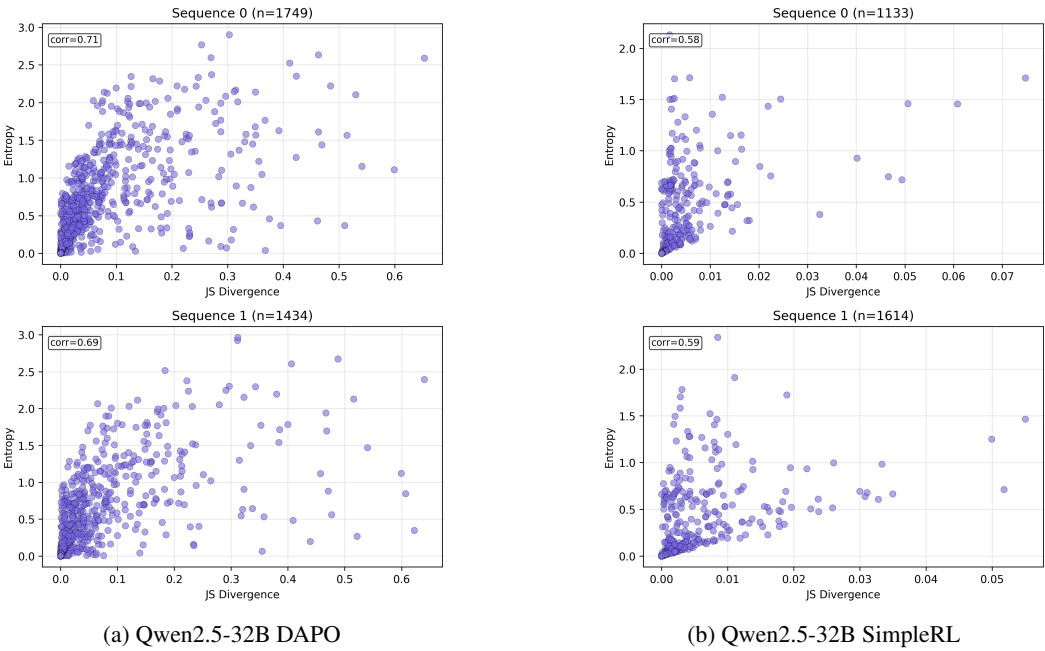

(a) Qwen2.5-32B DAPO

(b) Qwen2.5-32B SimpleRL

Figure 23: Per-sequence scatter plots relating entropy to JS divergence for Qwen2.5-32B DAPO and SimpleRL on AIME 2024. DAPO exhibits a broader entropy spread among divergent tokens, whereas SimpleRL concentrates divergence in higher-entropy regions.

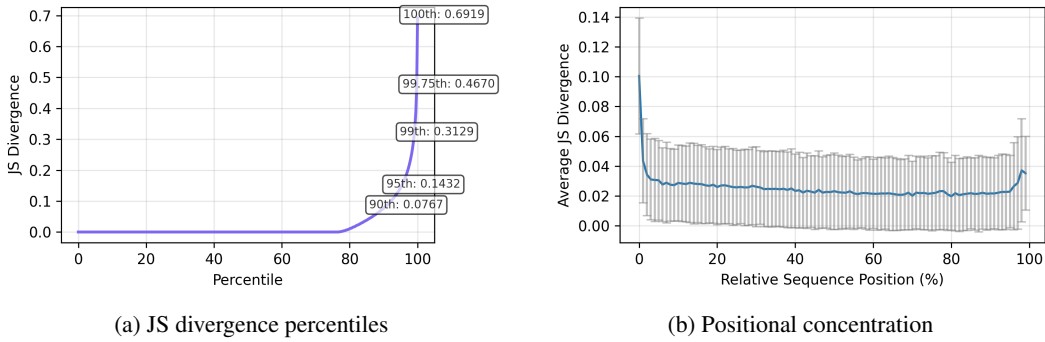

(a) JS divergence percentiles

(b) Positional concentration

Figure 24: JS divergence analysis for Qwen2.5-32B with DAPO on GPQA-Diamond. The sparsity patterns and positional concentration are consistent with findings on AIME datasets.

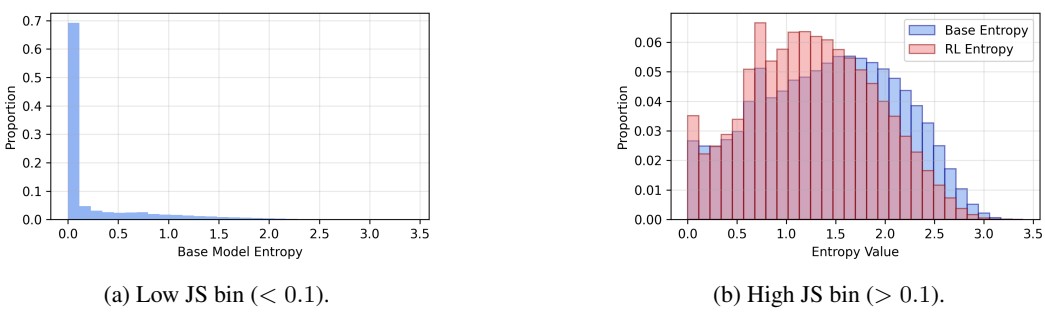

(a) Low JS bin ($< 0.1$).

(b) High JS bin ($> 0.1$).

Figure 25: Entropy distributions across divergence bins for Qwen2.5-32B with DAPO on GPQA-Diamond. Patterns are consistent with those observed on AIME datasets.

**Effect of Top-$p$ Sampling on JS Divergence.** To verify that our findings are robust to different top-$p$ sampling settings, we compare JS divergence distributions across different sampling configurations. The default setting uses top-$p = 0.7$ for sampling. We also evaluate configurations where sampling is performed with top-$p = 0.8$ and top-$p = 0.9$. Figure 26 shows that the sparsity patterns remain consistent across different sampling top-$p$ values, confirming that our results are not sensitive to the specific sampling top-$p$ value used.

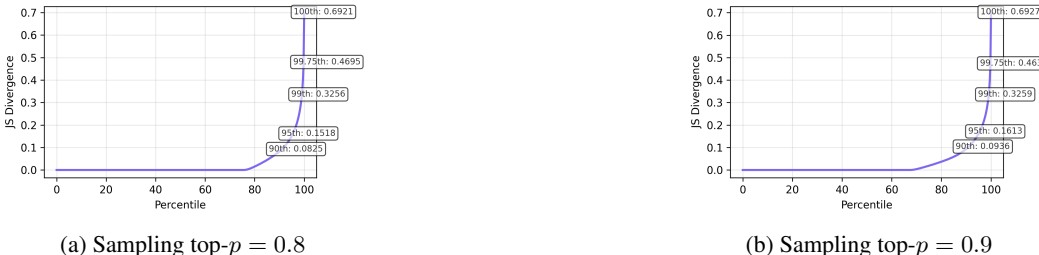

(a) Sampling top-$p = 0.8$                      (b) Sampling top-$p = 0.9$

Figure 26: JS divergence percentile curves for Qwen2.5-32B with DAPO on AIME 2024 under different top-$p$ sampling settings. The sparsity patterns remain consistent across different sampling top-$p$ values, indicating robustness to the specific sampling configuration.

**JS Divergence on AIME 2025.** Figure 27 shows JS divergence percentile curves for Qwen2.5-32B with DAPO and SimpleRL on AIME 2025, demonstrating consistent sparsity patterns across datasets.

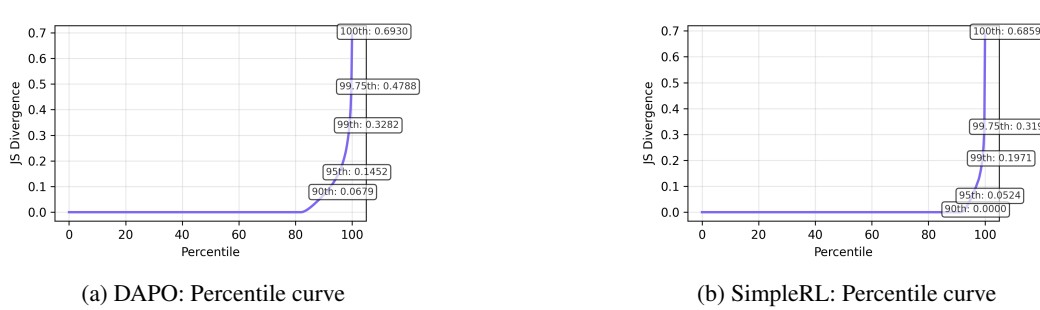

(a) DAPO: Percentile curve                    (b) SimpleRL: Percentile curve

Figure 27: JS divergence distributions for Qwen2.5-32B with DAPO and SimpleRL on AIME 2025. The sparsity patterns are consistent with those observed on AIME 2024, confirming the robustness of our findings across datasets.

**Effect of Top-$p$ Truncation on JS Divergence.** To verify that our use of top-$p$ truncated distributions (with topp $= 0.7$) does not significantly impact our findings, we compare JS divergence distributions computed using the estimated full distribution (top-$p = 1$) with those using truncated distributions. Figure 28 shows that the patterns remain consistent: distributional shifts are highly sparse regardless of truncation, with the vast majority of tokens showing near-zero divergence.

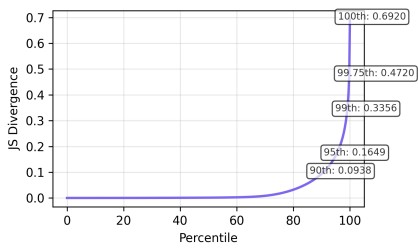

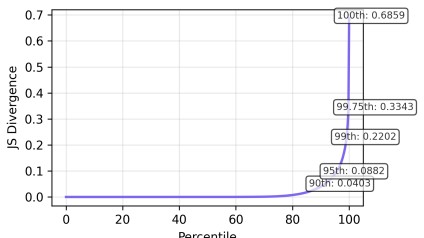

(a) DAPO: Percentile curve (topp1)

(b) SimpleRL: Percentile curve (topp1)

Figure 28: JS divergence distributions computed using top-$p = 1$ for Qwen2.5-32B with DAPO and SimpleRL on AIME 2025. The sparsity patterns are consistent with those observed using top-$p$ truncated distributions, confirming that truncation does not significantly impact our findings.

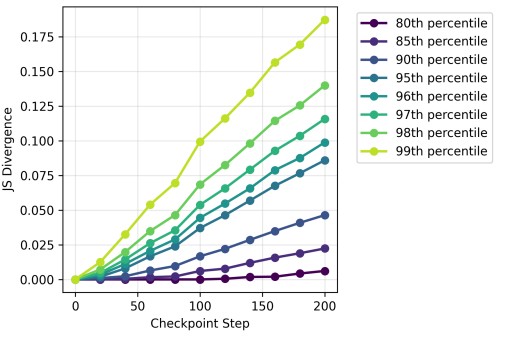

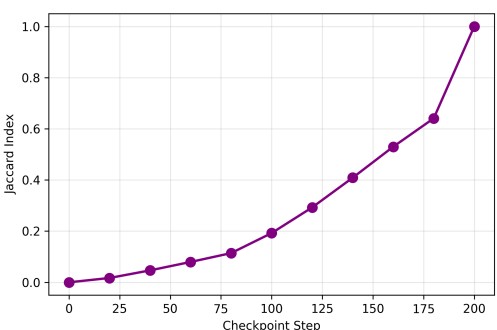

(a) JS divergence percentiles.

(b) Jaccard index with final divergent set ($JS_t > 0.1$).

Figure 29: Distributional shifts grow increasingly focused and stable. Most tokens remain unchanged; updates concentrate in a sparse set late in training.

### A.4.2 COMPARISON OF DAPO VARIANTS: CLIP-HIGHER SETTINGS

DAPO's clip-higher mechanism controls the degree of exploration during training. We compare two Qwen2.5-Math-7B models trained with DAPO: one with the default clip-higher setting (0.28) and another with a more restrictive setting (0.2). Figure 30 shows their JS divergence distributions on AIME 2024 and AIME 2025, revealing how the clip-higher parameter affects distributional shifts across datasets.

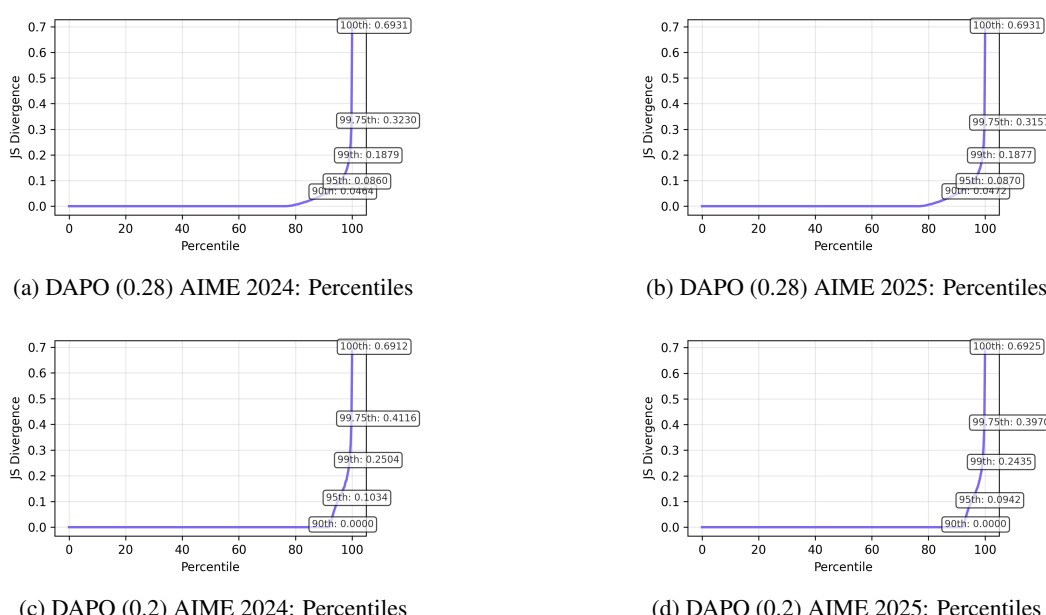

(a) DAPO (0.28) AIME 2024: Percentiles

(b) DAPO (0.28) AIME 2025: Percentiles

(c) DAPO (0.2) AIME 2024: Percentiles

(d) DAPO (0.2) AIME 2025: Percentiles

Figure 30: JS divergence distributions for Qwen2.5-Math-7B trained with DAPO under different clip-higher settings on AIME 2024 and AIME 2025. The more restrictive clip-higher=0.2 setting leads to sparser distributional shifts compared to the default 0.28 setting across both datasets, with a smaller proportion of tokens exhibiting nonnegligible divergence. However, on its divergent token set, the JS values are higher as indicated by the higher upper percentiles.

Figure 31 compares positional concentration patterns on AIME 2024 and AIME 2025, while Figure 32 and Figure 33 examine top-$k$ overlap and rank reordering, respectively. Figure 34 shows the percentage of divergent tokens whose RL top-1 choice had base probability below a given threshold for both DAPO variants across different datasets. Figure 35 shows entropy distributions across divergence bins for both DAPO variants.

Comparing the two upper clip variants (Figure 31), both clip settings exhibit larger average divergences at the beginning of the sequence, with a smaller increase near the end, consistent with the behavior seen in the 32B models. Interestingly, the 0.2 clip setting shows higher average divergence at the beginning of the sequence compared to the 0.28 setting.

For small $k$, clip-higher DAPO yields lower average overlap with the base model's top-$k$ set than the variant without clip-higher (Figure 32), indicating more frequent changes among the highest-ranked tokens. Interestingly, this trend reverses for larger $k$, where the model trained without clip-higher exhibits smaller overlap, suggesting that agreement with the base model deteriorates in the lower portion of the candidate set.

A consistent picture emerges from the rank-shift analysis in Figure 33. Without clip-higher, the base-model ranks of the RL model's top-3 tokens are more often preserved, but a non-negligible fraction of promoted tokens originate from much lower base ranks compared to the clip-higher variant. Taken together, these results suggest that clip-higher primarily redistributes probability mass within an already plausible candidate set, whereas removing clip-higher tends to strongly amplify a small number of existing top tokens. At the same time, some tokens promoted into the RL top ranks originate from much lower base ranks, particularly without clip-higher; however, given the lower

entropy observed without clip-higher, these tokens may still carry relatively small probability mass overall, with most probability concentrated on the top one or two tokens.

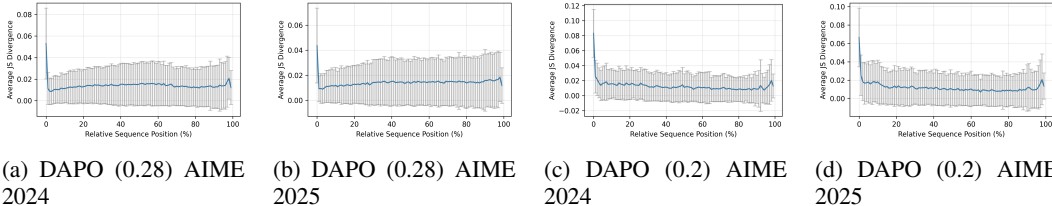

(a) DAPO (0.28) AIME 2024

(b) DAPO (0.28) AIME 2025

(c) DAPO (0.2) AIME 2024

(d) DAPO (0.2) AIME 2025

Figure 31: Mean JS divergence by normalized token position for DAPO variants with different clip-higher settings on AIME 2024 and AIME 2025.

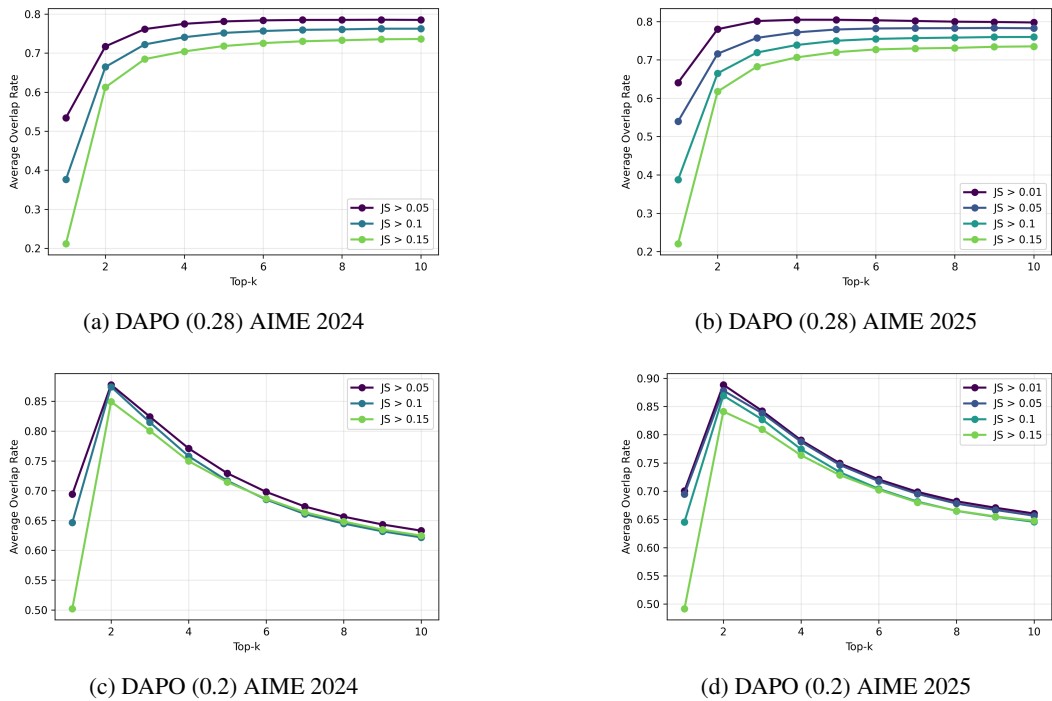

(a) DAPO (0.28) AIME 2024

(b) DAPO (0.28) AIME 2025

(c) DAPO (0.2) AIME 2024

(d) DAPO (0.2) AIME 2025

Figure 32: Top-$k$ token overlap between base and RL models at divergent positions ($\text{JS}_t > 0.1$) for DAPO variants on AIME 2024 and AIME 2025.

**Fine-tuning Data Results.** We also analyze distributional shifts on the fine-tuning data to examine how models behave on data they were fine-tuned on. Figure 36 shows JS divergence distributions, while Figures 37, 38 show additional analyses.

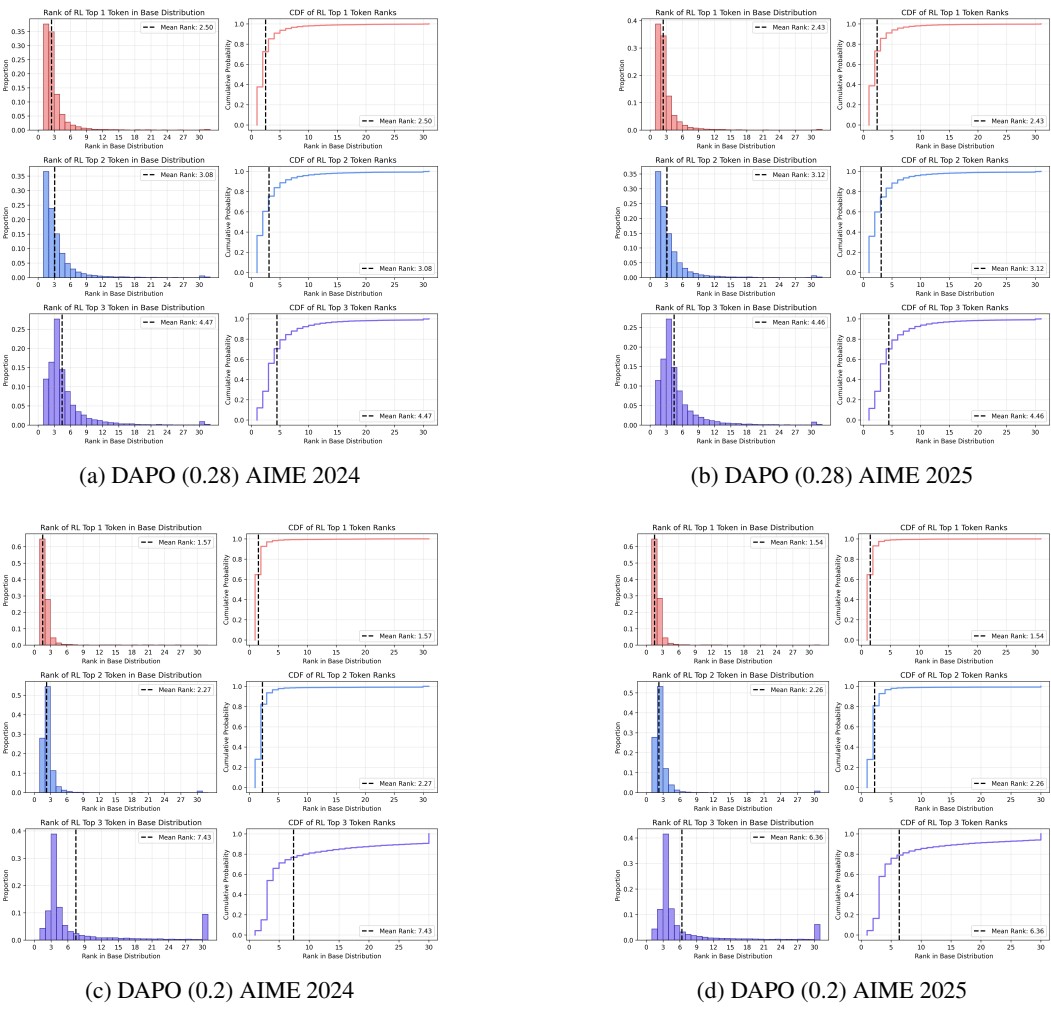

Figure 33: Distribution of base-model ranks for RL's top-3 tokens at high-divergence positions (JS > 0.1) for DAPO variants on AIME 2024 and AIME 2025.

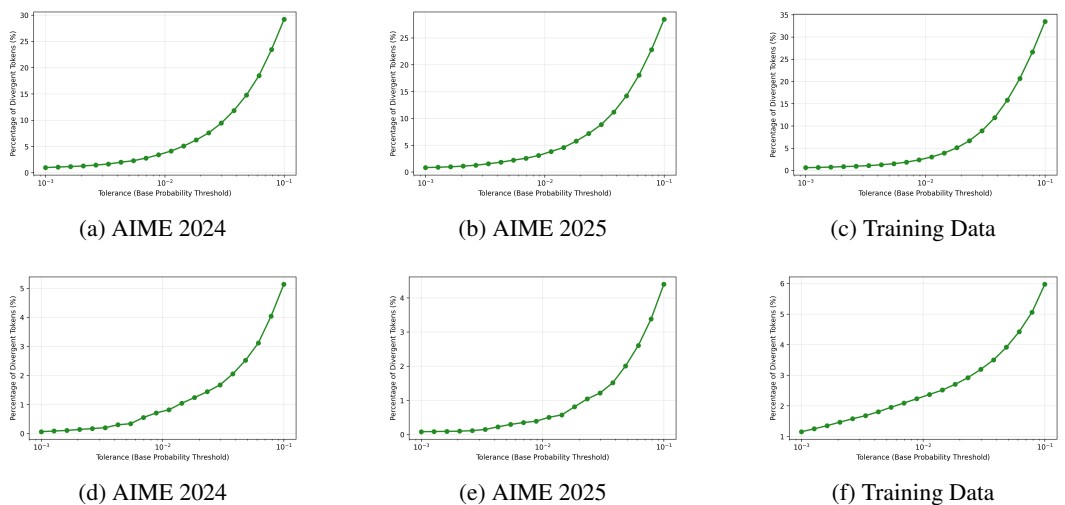

Figure 34: Percentage of divergent tokens whose RL top-1 choice had base probability below a given threshold for Qwen2.5-Math-7B with DAPO variants. Top row: DAPO (clip-higher=0.28); bottom row: DAPO (clip-higher=0.2). We further observe a distinction between the two clip-high settings, with the more restrictive setting (0.2) promoting substantially fewer tokens with low base probability.

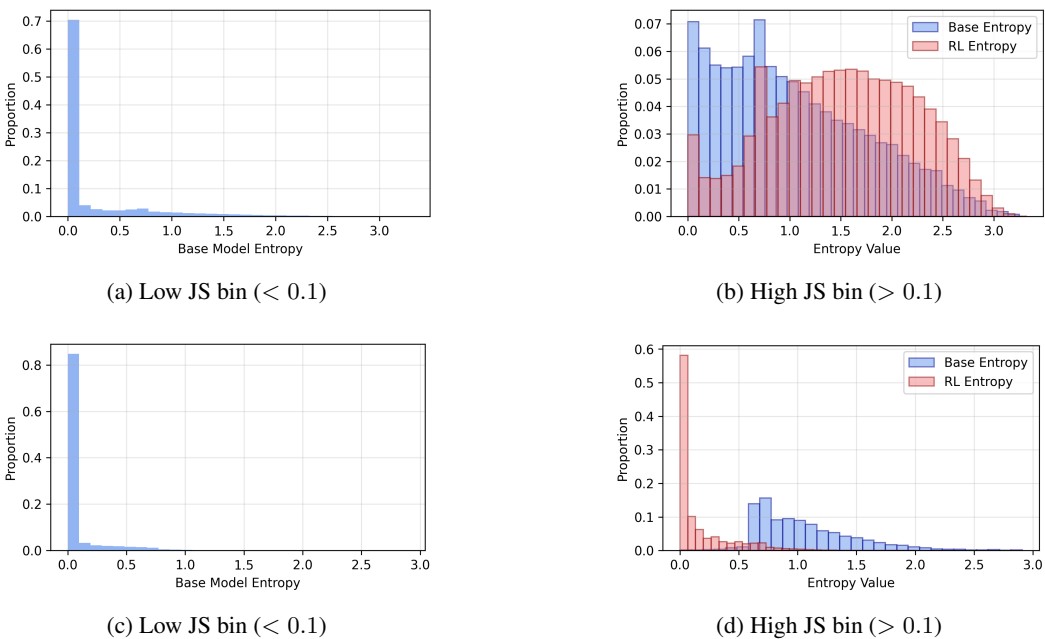

Figure 35: Entropy distributions across divergence bins for Qwen2.5-Math-7B with DAPO variants on AIME 2025. Top row: DAPO (clip-higher=0.28); bottom row: DAPO (clip-higher=0.2). Patterns are consistent with those observed in the main text, confirming the relationship between entropy and divergence across different clip-higher settings.

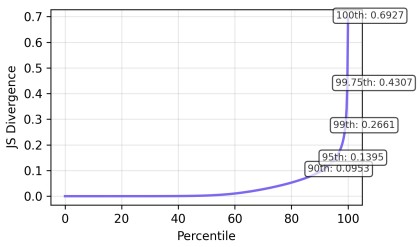

(a) DAPO (clip-higher=0.28): Percentiles       (b) DAPO (clip-higher=0.2): Percentiles

Figure 36: JS divergence distributions for DAPO variants of Qwen2.5-Math-7B on fine-tuning data (computed using approximated full distributions instead of truncated ones)

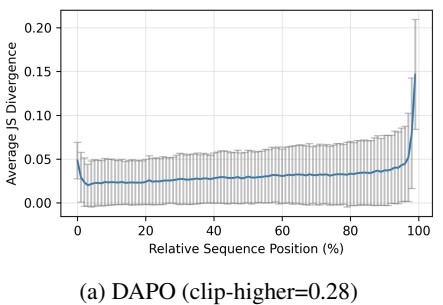
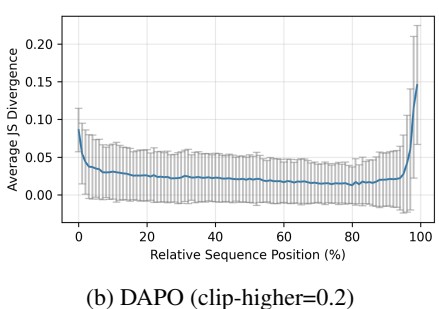

(a) DAPO (clip-higher=0.28)       (b) DAPO (clip-higher=0.2)

Figure 37: Mean JS divergence by normalized token position for DAPO variants on fine-tuning data.

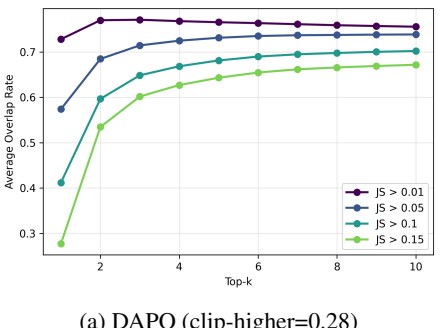
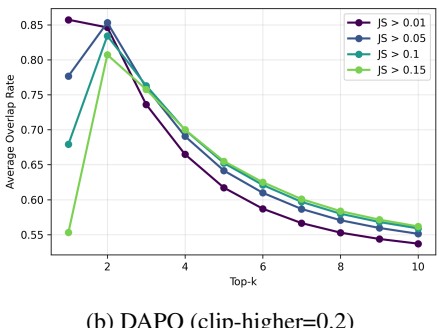

(a) DAPO (clip-higher=0.28)       (b) DAPO (clip-higher=0.2)

Figure 38: Top-$k$ token overlap between base and RL models at divergent positions ($\text{JS}_t > 0.1$) for DAPO variants on fine-tuning data.

### A.4.3 MISTRAL-SMALL-24B WITH SIMPLERL

We analyze Mistral-Small-24B trained with SimpleRL on AIME 2024 and AIME 2025 to demonstrate the generalizability of our findings across different model architectures. Figure 39 shows JS divergence percentile curves, revealing consistent sparsity patterns. Figure 40 shows positional concentration, Figure 41 shows entropy distributions across divergence bins, and Figure 42 shows tail behavior analysis.

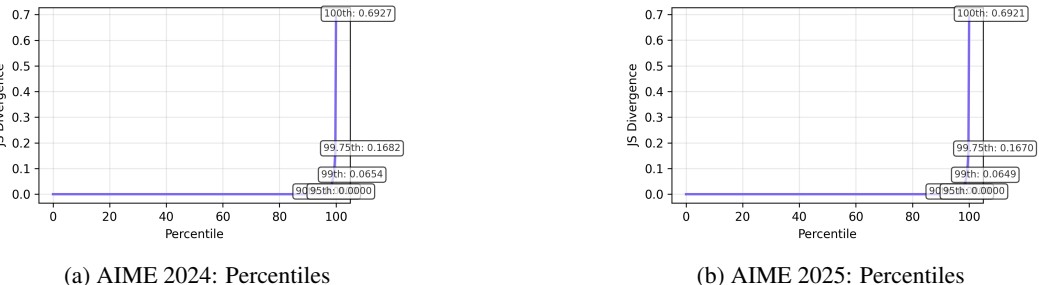

(a) AIME 2024: Percentiles      (b) AIME 2025: Percentiles

Figure 39: JS divergence distributions for Mistral-Small-24B with SimpleRL on AIME 2024 and AIME 2025. Sparse distributional shifts are consistent with findings in the main text across both datasets.

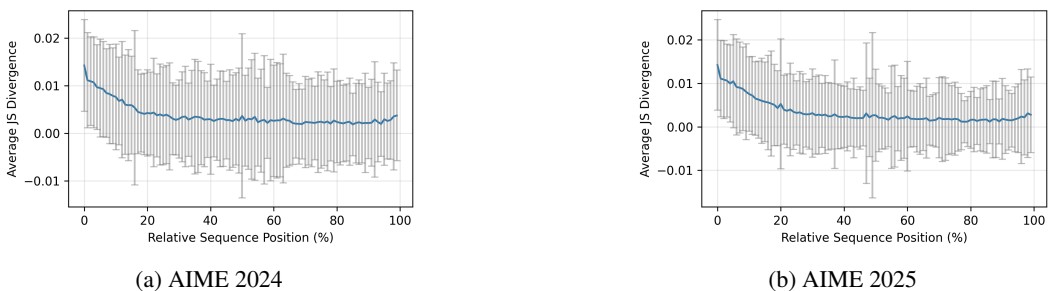

(a) AIME 2024      (b) AIME 2025

Figure 40: Mean JS divergence by normalized token position for Mistral-Small-24B with SimpleRL on AIME 2024 and AIME 2025. Consistent with findings for other models, average divergences are more concentrated at the start and end of responses.

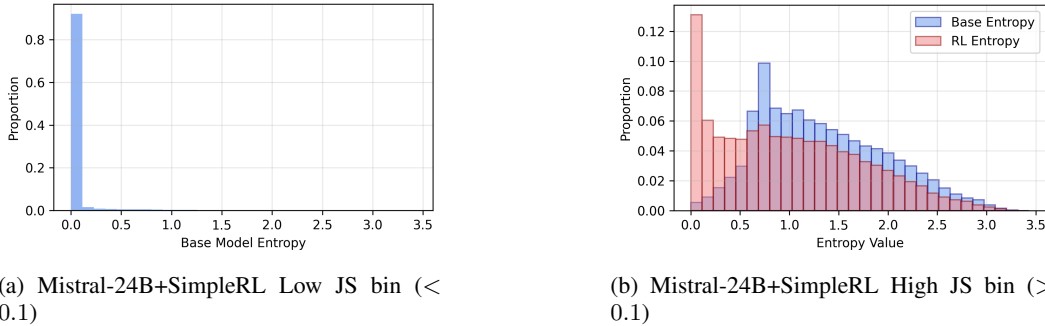

(a) Mistral-24B+SimpleRL Low JS bin ($<$ 0.1)      (b) Mistral-24B+SimpleRL High JS bin ($>$ 0.1)

Figure 41: Entropy distributions across divergence bins using full vocabulary for Mistral-Small-24B with SimpleRL on AIME 2024.

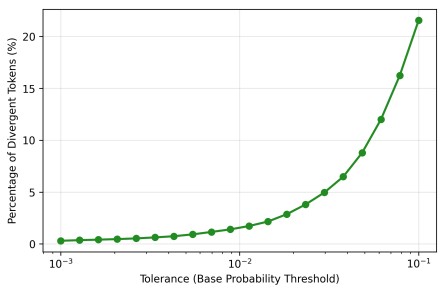
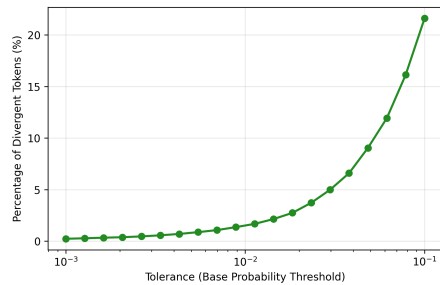

(a) Mistral-24B+SimpleRL AIME 2024       (b) Mistral-24B+SimpleRL AIME 2025

Figure 42: Percentage of divergent tokens whose RL top-1 choice had base probability below a given threshold for Mistral-Small-24B with SimpleRL on AIME 2024 and AIME 2025.

## A.5 ADDITIONAL CROSS-SAMPLING RESULTS AND DISCUSSION

This section provides supplementary cross-sampling results and the algorithm used for the cross-sampling experiments.

Algorithm 1 describes the general procedure for a single prompt (in pratice, we batch prompts for efficiency), which generates a response primarily under a *primary policy* and selectively intervenes using an *intervention policy* at positions where the token-level divergence exceeds a fixed threshold.

---

**Algorithm 1** Cross-Sampling for a Single Prompt

**Require:** Prompt prefix $x_{<1}$, primary policy $\pi_{\mathrm{prim}}$, intervention policy $\pi_{\mathrm{int}}$, divergence threshold $\epsilon_{\mathrm{JS}}$, maximum steps $T$
**Ensure:** Generated sequence $x_{1:t}$, number of intervention steps $k$
1: $k \leftarrow 0$
2: Initialize prefix $x_{<1}$
3: **for** $t = 1, \dots, T$ **do**
4:      Compute divergence $\mathrm{JS}_t = D_{\mathrm{JS}}(\pi_{\mathrm{prim}}(\cdot \mid x_{<t}) \parallel \pi_{\mathrm{int}}(\cdot \mid x_{<t}))$
5:      **if** $\mathrm{JS}_t > \varepsilon_{\mathrm{JS}}$ **then**
6:          Sample $x_t \sim \pi_{\mathrm{int}}(\cdot \mid x_{<t})$
7:          $k \leftarrow k + 1$
8:      **else**
9:          Sample $x_t \sim \pi_{\mathrm{prim}}(\cdot \mid x_{<t})$
10:      **end if**
11:      Append $x_t$ to prefix $x_{<t+1}$
12:      **if** $x_t = \mathrm{EOS}$ **then**
13:          **break**
14:      **end if**
15: **end for**
16: **return** generated sequence $x_{1:t}$ and intervention count $k$

---

**Progressive Steering of Reasoning Trajectories.** Across both forward and reverse cross-sampling, reasoning performance varies smoothly with the intervention budget. In the forward direction, accuracy improves steadily as more RL-sampled tokens are introduced, with no sharp threshold, indicating that gains accumulate across multiple sparse decision points rather than requiring all RL-induced changes. In reverse cross-sampling, performance degrades in a similarly smooth, near-monotonic manner as RL token choices are reverted, showing that RL-level performance depends on preserving a sparse set of token-level shifts throughout the trajectory. A key aspect in both settings is that interventions are applied sequentially, while decoding otherwise follows a single primary policy. One might expect that modifying only a few tokens, especially early ones, would have limited impact. However, this is not the case: **injecting just the first few RL tokens already yields measurable gains in forward cross-sampling, while reverting the earliest divergent tokens noticeably degrades performance in the reverse setting.** These effects arise because small local

edits can redirect the reasoning trajectory, which is then propagated by subsequent decoding under the primary policy.

**Connection to speculative decoding and BiLD.** Our cross-sampling framework is related to speculative decoding (Leviathan et al., 2023; Kim et al., 2023) in that generation depends on *two* next-token distributions. It is closer to *Big Little Decoder* (BiLD) (Kim et al., 2023), which defines a routing policy (rather than an exact sampling scheme that preserves a designated target distribution). BiLD trades off latency and quality via fallback and rollback rules, while our approach defines a mixed policy $\pi_{\text{mix}}^{(\text{prim,int})}$ to investigate the role of high-divergence decisions between base and RL models for reasoning performance. BiLD triggers fallback based on small-model confidence (e.g., a max-probability threshold) and uses rollback based on a discrepancy between small/large predictive distributions (with a cross-entropy–based quantity), whereas we intervene when $D_{\text{JS}}(\pi_{\text{prim}}(\cdot \mid X_{<t}) \parallel \pi_{\text{int}}(\cdot \mid X_{<t}))$ exceeds a threshold. BiLD further proposes prediction alignment by fine-tuning the small model on large-model outputs to reduce avoidable disagreements, while our setting typically uses models that already agree at most positions (base vs. RL) to isolate fine-tuning effects.

Table 3: Summary of cross-sampled tokens required to reach approximate RL-level performance (forward) or base-level performance (reverse) for Qwen2.5-32B on AIME 2024 and AIME 2025 with a token budget of 8000. Effective token counts/percentages exclude identity swaps during cross-sampling. Token percentages are computed at the sequence level.

| Dataset | Method | Eff. % Tokens | % Tokens | Eff. # Tokens | # Tokens | Initial Acc. | Final Acc. |
|---------|--------|---------------|----------|---------------|----------|--------------|------------|
| AIME24 | SimpleRL | 3.86% | 7.58% | 38 | 75 | 8.23 | > 25 |
| | SimpleRL Rev. | 5% | 8.3% | 29 | 51 | 25.52 | < 8.3 |
| | DAPO | 7.8% | 11.9% | 280 | 410 | 8.23 | > 44 |
| | DAPO Rev. | 10.1% | 14.9% | 173 | 258 | 44.8 | < 8.5 |
| AIME25 | SimpleRL | 1.53% | 2.97% | 13 | 26 | 5.3 | > 14 |
| | SimpleRL Rev. | 4.73% | 7.87% | 31 | 53 | 12.71 | < 4 |
| | DAPO | 6.47% | 9.18% | 230 | 326 | 4.8 | > 33 |
| | DAPO Rev. | 9.89% | 14.19% | 181 | 261 | 32 | < 4.5 |

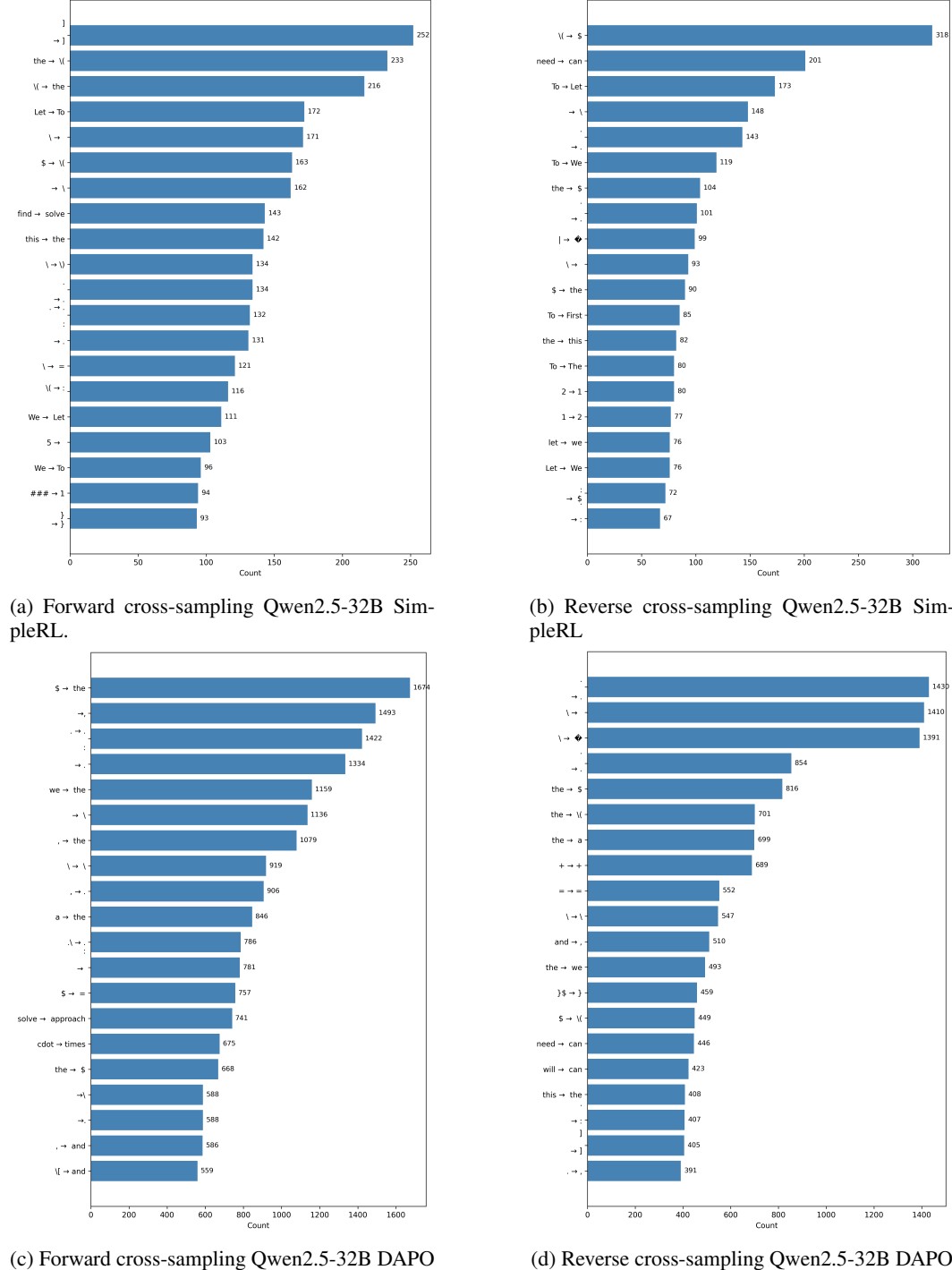

(a) Forward cross-sampling Qwen2.5-32B SimpleRL.

(b) Reverse cross-sampling Qwen2.5-32B SimpleRL

(c) Forward cross-sampling Qwen2.5-32B DAPO

(d) Reverse cross-sampling Qwen2.5-32B DAPO

Figure 43: Cross-sampling token pair histograms.

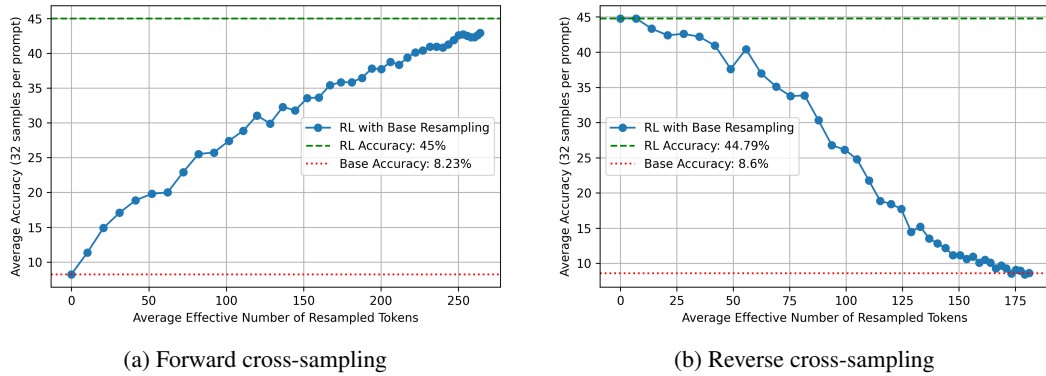

(a) Forward cross-sampling

(b) Reverse cross-sampling

Figure 44: Cross-sampling results (DAPO on AIME 2024): injecting RL tokens into base generations progressively recovers RL accuracy, while reverting RL tokens with base tokens causes near-monotonic degradation toward base performance.

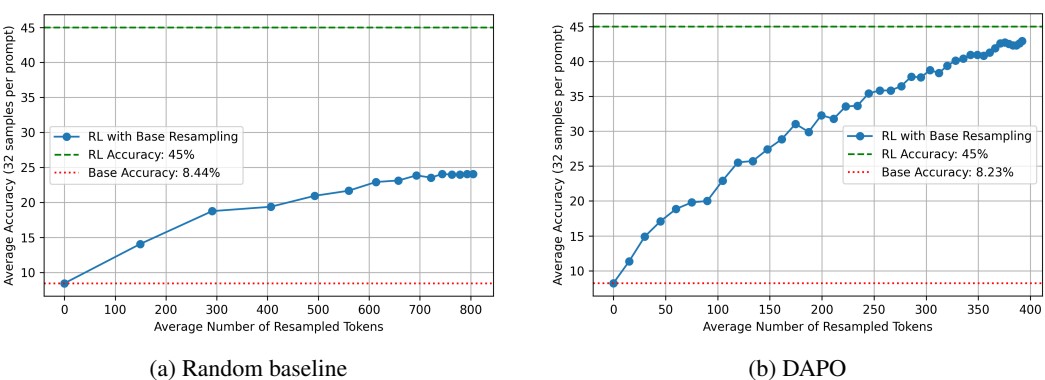

(a) Random baseline

(b) DAPO

Figure 45: Comparison of random baseline and DAPO cross-sampling on AIME 2024: average number of tokens (including identity swaps) replaced versus accuracy. The random baseline shows minimal performance improvement, demonstrating that targeted RL token selection is critical for performance gains.

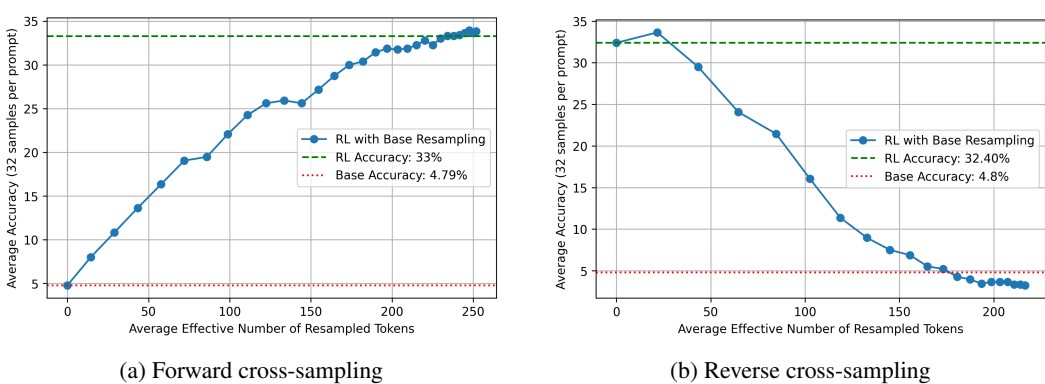

(a) Forward cross-sampling

(b) Reverse cross-sampling

Figure 46: Cross-sampling results (DAPO on AIME 2025): injecting RL tokens into base generations progressively recovers RL accuracy, while reverting RL tokens with base tokens causes near-monotonic degradation toward base performance.

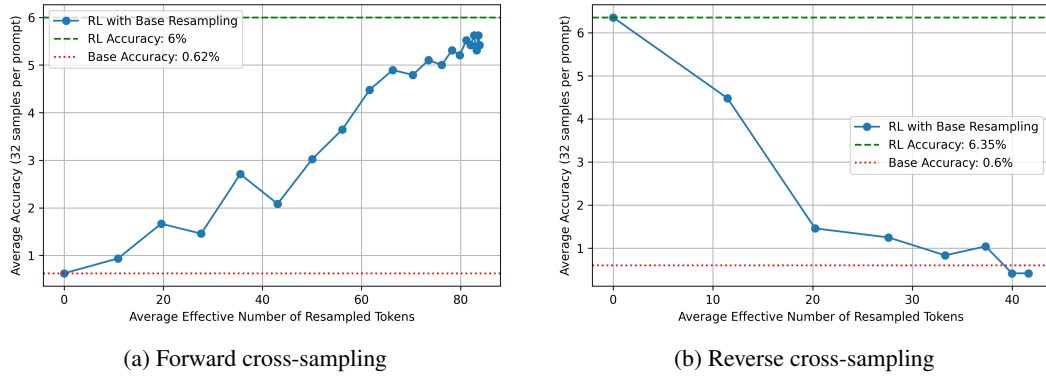

(a) Forward cross-sampling

(b) Reverse cross-sampling

Figure 47: Cross-sampling results (Mistral-Small-24B + SimpleRL on AIME 2024): injecting RL tokens into base generations progressively recovers RL accuracy, while reverting RL tokens with base tokens causes near-monotonic degradation toward base performance.

### A.6 SEQUENCE-LEVEL DIVERGENCE BOUNDS FOR CROSS-SAMPLING

**Setup.** Let $(X_t)_{t\geq 1}$ be the decoded tokens (each in $\mathcal{V}$), with stopping time $\tau := \inf\{t \geq 1 : X_t = \text{EOS}\} \wedge T_{\max}$. We may work on the fixed horizon $T_{\max}$ by absorbing the EOS token: once EOS is generated, the process deterministically outputs EOS thereafter. This yields an equivalent distribution over $X_{1:T_{\max}}$ and ensures all sequences are of length $T_{\max}$. All results below therefore sum over $t = 1, \ldots, T_{\max}$; terms after $\tau$ contribute zero since both policies become point masses on EOS.

Let $\pi_{\text{prim}}$ be the primary policy and $\pi_{\text{int}}$ the intervention policy. Given a switching rule $\mathcal{S} : \mathcal{V}^{<\mathbb{N}} \to \{0, 1\}$, define $S_t := \mathcal{S}(X_{<t})$ and the mixed policy

$$\pi_{\text{mix}}(\cdot \mid X_{<t}) = (1 - S_t)\,\pi_{\text{prim}}(\cdot \mid X_{<t}) + S_t\,\pi_{\text{int}}(\cdot \mid X_{<t}).$$

Let $P_{\text{mix}}$ and $P_{\text{int}}$ denote the induced sequence-level distributions on $X_{1:T_{\max}}$.

We provide a bound on the sequence-level divergence between the cross-sampled policy and the target intervention policy, in the simpler case of KL divergence with a token-level KL switching rule.

**Lemma A.1** (KL decomposition). *Let $P, Q$ be distributions on $X_{1:T_{\max}}$ admitting factorizations $P(x_{1:T_{\max}}) = \prod_{t=1}^{T_{\max}} p_t(x_t \mid x_{<t})$ and $Q(x_{1:T_{\max}}) = \prod_{t=1}^{T_{\max}} q_t(x_t \mid x_{<t})$. Then*

$$D_{\text{KL}}(P \parallel Q) = \sum_{t=1}^{T_{\max}} \mathbb{E}_{X_{<t} \sim P}\big[D_{\text{KL}}\big(p_t(\cdot \mid X_{<t}) \parallel q_t(\cdot \mid X_{<t})\big)\big].$$

*Proof.* By definition, $D_{\text{KL}}(P \parallel Q) = \mathbb{E}_{X \sim P}\left[\log \frac{P(X)}{Q(X)}\right]$. Using the factorizations,

$$\log \frac{P(X_{1:T_{\max}})}{Q(X_{1:T_{\max}})} = \sum_{t=1}^{T_{\max}} \log \frac{p_t(X_t \mid X_{<t})}{q_t(X_t \mid X_{<t})}.$$

Taking expectation under $P$ and exchanging sum and expectation gives

$$\begin{aligned}
D_{\text{KL}}(P \parallel Q) &= \sum_{t=1}^{T_{\max}} \mathbb{E}_{X \sim P}\left[\log \frac{p_t(X_t \mid X_{<t})}{q_t(X_t \mid X_{<t})}\right] \\
&= \sum_{t=1}^{T_{\max}} \mathbb{E}_{X_{<t} \sim P}\left[\mathbb{E}_{X_t \sim p_t(\cdot \mid X_{<t})}\left[\log \frac{p_t(X_t \mid X_{<t})}{q_t(X_t \mid X_{<t})}\right]\right] \\
&= \sum_{t=1}^{T_{\max}} \mathbb{E}_{X_{<t} \sim P}\big[D_{\text{KL}}\big(p_t(\cdot \mid X_{<t}) \parallel q_t(\cdot \mid X_{<t})\big)\big].
\end{aligned}$$

where the second last equality follows by the definition of conditional expectation (or the law of total expectation) applied to the conditional distribution of $X_t$ given $X_{<t}$. $\qquad\square$

**Proposition A.2** (Token-level KL threshold $\Rightarrow$ sequence-level KL bound). *Assume the switching rule is defined by a KL threshold:*

$$\mathcal{S}(x_{<t}) = \mathbb{1}\big\{D_{\mathrm{KL}}\big(\pi_{\mathrm{prim}}(\cdot \mid x_{<t}) \,\|\, \pi_{\mathrm{int}}(\cdot \mid x_{<t})\big) > \varepsilon\big\}.$$

*Define the number of* non-intervention *steps on a trajectory*

$$N_0 := \sum_{t=1}^{\tau} \mathbb{1}\{\mathcal{S}(X_{<t}) = 0\}.$$

*Then*

$$D_{\mathrm{KL}}(P_{\mathrm{mix}} \,\|\, P_{\mathrm{int}}) \le \varepsilon\, \mathbb{E}_{X \sim P_{\mathrm{mix}}}[N_0].$$

*Proof.* We apply Lemma A.1 with $P = P_{\mathrm{mix}}$ and $Q = P_{\mathrm{int}}$. For any history $h = x_{<t}$, since $\mathcal{S}(h) \in \{0, 1\}$,

$$\pi_{\mathrm{mix}}(\cdot \mid h) = \begin{cases} \pi_{\mathrm{int}}(\cdot \mid h), & \mathcal{S}(h) = 1, \\ \pi_{\mathrm{prim}}(\cdot \mid h), & \mathcal{S}(h) = 0. \end{cases}$$

Hence

$$D_{\mathrm{KL}}\big(\pi_{\mathrm{mix}}(\cdot \mid h) \,\|\, \pi_{\mathrm{int}}(\cdot \mid h)\big) = \mathbb{1}\{S(h) = 0\}\, D_{\mathrm{KL}}\big(\pi_{\mathrm{prim}}(\cdot \mid h) \,\|\, \pi_{\mathrm{int}}(\cdot \mid h)\big).$$

By the definition of $\mathcal{S}$, whenever $\mathcal{S}(h) = 0$ the token-level KL is at most $\varepsilon$. Therefore, for all $h$,

$$D_{\mathrm{KL}}\big(\pi_{\mathrm{mix}}(\cdot \mid h) \,\|\, \pi_{\mathrm{int}}(\cdot \mid h)\big) \le \varepsilon\, \mathbb{1}\{\mathcal{S}(h) = 0\}.$$

Under absorbing EOS, $D_{\mathrm{KL}}\big(\pi_{\mathrm{mix}}(\cdot \mid X_{<t}) \,\|\, \pi_{\mathrm{int}}(\cdot \mid X_{<t})\big) = 0$ for $t > \tau$, so we may insert $\mathbb{1}\{t \le \tau\}$ in the sum. Substituting into Lemma A.1 with this yields

$$D_{\mathrm{KL}}(P_{\mathrm{mix}} \,\|\, P_{\mathrm{int}}) \le \sum_{t=1}^{T_{\max}} \mathbb{E}_{X_{<t} \sim P_{\mathrm{mix}}}[\mathbb{1}\{t \le \tau\}\, \varepsilon\, \mathbb{1}\{S_t = 0\}] = \varepsilon \sum_{t=1}^{T_{\max}} \mathbb{E}_{X_{<t} \sim P_{\mathrm{mix}}}[\mathbb{1}\{t \le \tau\}\, \mathbb{1}\{S_t = 0\}].$$

Note that $\{t \le \tau\} = \{\text{EOS} \notin X_{<t}\}$ depends only on $X_{<t}$, so $\mathbb{1}\{t \le \tau\}\mathbb{1}\{S_t = 0\}$ is measurable with respect to $\sigma(X_{<t})$. Thus, we may equivalently write the expectation under the full trajectory $X \sim P_{\mathrm{mix}}$:

$$\mathbb{E}_{X_{<t} \sim P_{\mathrm{mix}}}[\mathbb{1}\{t \le \tau\}\, \mathbb{1}\{S_t = 0\}] = \mathbb{E}_{X \sim P_{\mathrm{mix}}}[\mathbb{1}\{t \le \tau\}\, \mathbb{1}\{S_t = 0\}].$$

Then,

$$\varepsilon \sum_{t=1}^{T_{\max}} \mathbb{E}_{X_{<t} \sim P_{\mathrm{mix}}}[\mathbb{1}\{t \le \tau\}\, \mathbb{1}\{S_t = 0\}] = \varepsilon \sum_{t=1}^{T_{\max}} \mathbb{E}_{X \sim P_{\mathrm{mix}}}[\mathbb{1}\{t \le \tau\}\, \mathbb{1}\{S_t = 0\}]$$

$$= \varepsilon\, \mathbb{E}_{X \sim P_{\mathrm{mix}}}\left[\sum_{t=1}^{T_{\max}} \mathbb{1}\{t \le \tau\}\, \mathbb{1}\{S_t = 0\}\right]$$

$$= \varepsilon\, \mathbb{E}_{X \sim P_{\mathrm{mix}}}\left[\sum_{t=1}^{\tau} \mathbb{1}\{S_t = 0\}\right]$$

$$= \varepsilon\, \mathbb{E}_{X \sim P_{\mathrm{mix}}}[N_0].$$

$\qquad\square$

*Remark* A.3 (Effective KL on non-intervention steps). Define the *effective* token-level KL on non-intervention steps

$$\bar{\kappa} := \frac{\mathbb{E}_{X \sim P_{\mathrm{mix}}}\left[\sum_{t=1}^{\tau} \mathbb{1}\{S_t = 0\}\, D_{\mathrm{KL}}\big(\pi_{\mathrm{prim}}(\cdot \mid X_{<t}) \,\|\, \pi_{\mathrm{int}}(\cdot \mid X_{<t})\big)\right]}{\mathbb{E}_{X \sim P_{\mathrm{mix}}}[N_0]}, \qquad (\bar{\kappa} := 0 \text{ if } \mathbb{E}[N_0] = 0).$$

Then

$$D_{\mathrm{KL}}(P_{\mathrm{mix}} \,\|\, P_{\mathrm{int}}) = \bar{\kappa}\, \mathbb{E}_{X \sim P_{\mathrm{mix}}}[N_0] \le \varepsilon\, \mathbb{E}_{X \sim P_{\mathrm{mix}}}[N_0],$$

and typically $\bar{\kappa} \ll \varepsilon$ at most steps in regimes where the models are already close (eg. in the setting of a base model and its RL fine-tuned counterpart).

## A.7 ADDITIONAL DIVERGENCE-WEIGHTED ADVANTAGE DISCUSSION/RESULTS

This section presents supplementary results for divergence-weighted advantages, including alternative configurations and evaluations on Qwen2.5-7B. The main experiments focus on Qwen2.5-Math-7B with KL divergence computed with respect to $\pi_{\theta_{\text{old}}}$ and sigmoid weighting. Here we discuss alternative schemes, along with additional results.

### A.7.1 ALTERNATIVE DIVERGENCE CHOICES

Beyond the old-policy KL divergence presented in the main text, one could also use the reference-based KL divergence:

$$\text{KL}_t^{\text{ref}} = D_{\text{KL}}(\pi_{\theta_{\text{old}}}(\cdot \mid x_{<t}) \parallel \pi_{\text{ref}}(\cdot \mid x_{<t})), \tag{6}$$

where $\pi_{\text{ref}}$ denotes the base reference model. The reference-based KL quantifies the alignment between the current policy and the original base model, measuring the cumulative divergence from the initial model at each token position. This contrasts with the old-policy KL, which captures only the magnitude of recent policy updates within a single training iteration. In our experiments we mainly just use the old-policy KL as this does not require additional pass through the base model.

### A.7.2 ALTERNATIVE WEIGHTING SCHEMES

In addition to the sigmoid weighting scheme presented in the main text, we examine linear relative weighting:

$$w_t = 1 + \alpha(\text{KL}_t - \mu_{\text{KL}}), \quad \mu_{\text{KL}} = \tfrac{1}{T}\sum_{t=1}^{T} \text{KL}_t. \tag{7}$$

The linear relative scheme scales weights linearly with the deviation from the mean KL divergence across the sequence, offering a simpler alternative to the sigmoid transformation. As in the sigmoid case, $\alpha > 0$ amplifies high-divergence tokens, while $\alpha < 0$ emphasizes low-divergence ones.

### A.7.3 RESULTS ON ADDITIONAL CONFIGURATIONS

Table 4 summarizes the performance of these alternative configurations, including evaluations using linear relative weighting on Qwen2.5-7B.

Table 4: Accuracy (%) under additional divergence-weighted configurations on Qwen2.5-7B. Results shown across AIME 2024, AIME 2025, and AMC datasets. The results displayed are the avg@32 scores.

| Configuration | AIME24 | AIME25 | AMC | Overall Avg |
|---|---|---|---|---|
| Baseline DAPO | 16.77 | 8.12 | 70.78 | 31.89 |
| High-KL Lin. Rel. (sched.) | 19.58 | 12.40 | 71.12 | 34.37 |
| High-KL Lin. Rel. | 20.00 | 12.29 | 73.31 | 35.20 |

Table 5: Accuracy (%) for 80/20 clip entropy configuration on Qwen2.5-Math-7B. Results shown across AIME 2024, AIME 2025, and AMC datasets. The results displayed are the avg@32 scores.

| Configuration | AIME24 | AIME25 | AMC | Overall Avg |
|---|---|---|---|---|
| 80/20 clip entropy | 35.26 | 17.03 | 72.68 | 41.66 |

## A.8 WEIGHT-LEVEL ANALYSIS OF CHANGES

Orthogonal to the analysis done in the main text, we also investigate the degree of modifications induced by RLVR at the parameter level. More specifically, we employ the relative gap ratio (Wu et al., 2025), denoted as $\sigma$, to quantify the magnitude of weight divergence pre- and post-fine-tuning. This ratio is formulated as:

$$\sigma = \frac{\sum |W_{\text{original}} - W_{\text{tuned}}|}{\sum |W_{\text{original}}| + \sum |W_{\text{tuned}}|}$$

where $W_{\text{original}}$ and $W_{\text{tuned}}$ represent the model parameters before and after fine-tuning, respectively. A lower $\sigma$ value signifies greater similarity between the parameter sets, indicating a smaller overall modification from the fine-tuning process.

In our experiment, we utilized the Qwen2.5-32B and Qwen2.5-Math-7B models as foundations. Each model was independently fine-tuned via two distinct methodologies: RL and Supervised Fine-Tuning (SFT). To ensure a controlled and equitable comparison, the training regimen for both methods was standardized, employing an identical dataset size and the same number of training steps. Subsequently, the $\sigma$ was computed between each original model and its corresponding tuned counterparts. The results are presented in the following table.

Table 6: Relative gap ratio ($\sigma$) after RL and SFT fine-tuning.

| Model | Qwen2.5-32B | Qwen2.5-Math-7B |
|---|---|---|
| $\sigma$ after RL | 0.00143 | 0.00136 |
| $\sigma$ after SFT | 0.00347 | 0.00944 |

The results presented in the table demonstrate a consistent trend across both models: the $\sigma$ values corresponding to RL fine-tuning are substantially lower than those from SFT. This quantitative analysis at the parameter level suggests that the cumulative weight modifications induced by RL are significantly less extensive than those resulting from SFT. This finding provides empirical support for the hypothesis that RL achieves performance gains through sparse and targeted parameter adjustments, contrasting with the more distributed updates characteristic of SFT.

## A.9 LLM USAGE

LLMs were used mainly to assist with minor polishing and organizing of the writing.

