# OpenReview forum: "Sparse but Critical: A Token-Level Analysis of Distributional Shifts in RLVR Fine-Tuning of LLMs"
_ICLR.cc/2026/Conference — ICLR 2026 Poster_

### Official Review · Reviewer_SHpG · 2025-10-30

**Soundness:** 3
**Presentation:** 3
**Contribution:** 2
**Rating:** 4
**Confidence:** 3

**Summary:**

This paper studies the changes RLVR introduces by analyzing the token-level divergences between pre-trained and RLVR-trained versions of the Qwen2.5-7B model. The authors conclude that changes are concentrated in a sparse set of high divergence tokens, and conduct several experiments to validate and explain this finding: the cross-sampling analysis where high-divergence tokens are sampled from RLVR-trained model and put into the base model, and vice versa; the semantic analysis of changed tokens; the dynamics of changes during training.

**Strengths:**

S1. The results are novel to my knowledge and are presented in a concise manner; analysis in the paper is well-motivated; paper is dense and filled with results that generally support their claims and conclusions. Main claims of the paper are supported, but the execution of experiments might be significantly strenghtened further (see weaknesses).

S2. Experiments with cross-sampling and advantage reweighting are interesting and clever and seem to be a promising analysis toolkit; however, see W3 and W4.

**Weaknesses:**

W1. The most serious limitation of the work is that it lacks the treatment of other post-training schemes such as SFT, preference optimization or RLHF, so it is not clear if conclusions made in paper are specific to RLVR and not to fine-tuning in general.

W2. It is well-known that Qwen family shows different properties when training with RLVR on mathematics [1], and the paper does not study other families and tasks. Although these results are recent and the authors might be not aware about them when preparing the manuscript, this limits the generalisability of the paper findings and should be acknowledged in future work.

W3. All comparisons between GRPO and DAPO are affected by the different datasets the studied models were trained on, and this factor is not isolated. This is acknowledged in a paper.

W4. For experiment with cross-sampling, an important control baseline is missing - for example, the case when a token is sampled from another model randomly, instead of divergence-based rule, or something of that kind. This will quantify the effect of high-divergence replacements, and if control baseline would give lower final accuracy, then the conclusion made in line 268 "the improvements from RL fine-tuning are concentrated in a sparse set of high divergence tokens" would be reliable.

W5. Experiment with advantage reweighting in section 6.2 is underdeveloped, shows minor improvement, and it is unclear if the method allows for better results and stable training; another major shortcoming is that only DAPO baseline is analyzed. Since reweighting algorithm appears to use KL divergence, it is unclear how would this algorithm work if baseline algorithm already have KL divergence penalty (e.g. GRPO). Overall, this section is not very useful for the reader and further research in its current form, but the method seems interesting.

W6. As a minor weakness, although paper has a lot of results presented in a concise manner, it is densely written and it is slightly hard to parse and understand the meaning and interpretation of some results. I think that moving some content from main body to appendix and focusing on more clear presentation would improve the paper, but this is more suited for further iterations of the work.

**Questions:**

Q1. In figures 3 and 9, in the low JS bin, am I correct that RL entropy (red) almost zero and therefore is not visible? Please correct me if I misunderstood the section 3.3. Perhaps, the scatterplot or 2D histogram with JS / entropy axes would be effective; it also might be made for each sequence separately, and each point would represent separate token in a sequence.

Q2. Have you conducted an experiment where position would be measured not relative to the end of the sequence, but to specific elements of it (e.g. distance from system prompt, from the instruction, model generation start, from the final answer to the question etc.)? If yes, then what have you noticed? This is partially addressed by section 3.4; since the result in figure 10 suggest that single token might operate in low-divergence and high-divergence regimes, the positional information might reveal when some token would have high entropy and high divergence.

Q3. Is it possible to extend percentage of replaced tokens in the figure 5? Would trend be kept or will it be bounded from above by RL accuracy and from below by base model accuracy?

Q4. Results in section 5.1 are shown for high-divergence positions and therefore we expect differences to be noticeable; what changes if we do not restrict the positions and calculate this ranking results for all tokens? I also suggest to make shared x-axis for columns in figure 11 (b) so it would be easier to compare.

Q5. Section 3.2 about the dependency of entropy shift on position in a sequence is not substantive enough: absolute values of JS divergences shown for both methods are very low, and trend is seen only for DAPO. Also, since for DAPO the values are significantly larger than for GRPO (SimpleRL) as in section 3.1, it suggests that entropy shift might be the consequence of nonzero KL divergence regularization term and similar mechanisms to preserve the generation behaviour. Can you speculate if this is the case, or correct me if I'm wrong?

Q6. Have you gathered more results on advantage reweighting experiments? Currently, improvement seems very small. Have you identified for which scenarios and baseline algorithms this method is meaningfully applicable?

Considering cross-sampling experiment, you might be interested in checking out [2].

[1] Spurious Rewards: Rethinking Training Signals in RLVR, Shao et al.
[2] Reasoning with Sampling: Your Base Model is Smarter Than You Think, Karan A., Du Y., 2025.

---

> ### Author Response · Authors · 2025-11-24
>
> We thank the reviewer for the extensive and thoughtful feedback. Below we address the main weaknesses and questions raised by the reviewer.
>
> ### (A) Generality Beyond RLVR (SFT, Preference Methods, Other Families, Other Tasks)
> > **W1:** “The most serious limitation is that the work lacks the treatment of other post-training schemes such as SFT, preference optimization or RLHF… it is not clear if the conclusions are specific to RLVR.”
>
> We appreciate the reviewer’s concern regarding whether our conclusions are specific to RLVR or hold more broadly across post-training methods. To address this, we have extended our analysis to include a Supervised Fine-Tuning (SFT) model based on Qwen2.5-32B, presented in Appendix A.2. This new experiment allows us to directly compare the structural and distributional behaviors of SFT and RLVR under the same evaluation settings. Our results show that SFT exhibits notably different behavior from RLVR across, for example
> - Higher average JS divergence and a larger set of high-divergence tokens, indicating that SFT modifies the model in a more globally distributed manner rather than concentrating changes as RLVR does.
> - Substantially lower top-k overlap between base and SFT distributions, suggesting that SFT shifts the ranking structure more aggressively, whereas RLVR preserves most of the top token overlap, even among high-divergence distributions.
> - Much larger rank shifts across tokens, further highlighting that SFT introduces broader, less targeted changes.
> - A higher proportion of tokens with low base probability receiving elevated mass in the SFT model’s distribution, again contrasting with RLVR’s targeted elevation of probabilities for a small set of critical tokens.
>
> Together, these findings show that RLVR’s behavior is indeed distinct from SFT from the perspective of our analysis. This supports our claim that the structural and distributional patterns identified in the paper, such as sparsity of shifts, are not generic consequences of fine-tuning.
>
>
> ### (B) Generalization Across Model Families and Non-Math Tasks
> > **W2:** “It is well-known that Qwen shows different properties… and the paper does not study other families and tasks.”
>
> We thank the reviewer for highlighting this line of work [1]; we have added the cited reference to the manuscript. To address generalizability, we have extended our experiments beyond Qwen. In particular, we include results for Mistral Small 24B trained with SimpleRL (Appendix A.4.3), and we observe very similar structural patterns, including sparsity of shifts, as well as through the cross-sampling experiments. We also include additional experiments on the GPQA benchmark, which spans multiple reasoning domains, beyond math-specific data.
>
> [1] Spurious Rewards: Rethinking Training Signals in RLVR, Shao et al.
>
> ### (C) Dataset Confounds in GRPO vs. DAPO Comparisons
> > **W3:** “Comparisons between GRPO and DAPO are confounded by differing training datasets.”
>
> We agree with the reviewer that comparisons between GRPO and DAPO may be influenced by differences in the underlying training data, especially at the 32B scale where we rely on publicly released reasoning models. We acknowledge this limitation in the manuscript.
>
> To further isolate the effect of the algorithmic differences, we additionally report results on Qwen2.5-Math 7B trained with the DAPO recipe on the same data and for the same number of steps. The only controlled variation we introduce is the clip-higher value, using the standard DAPO setting of 0.28 versus 0.2. The behaviors we observe align with the expectations of less "exploration" under the 0.2 setting (Appendix A.4.2): namely sparser divergence patterns, stronger RL entropy collapse, fewer low-base-entropy tokens appearing in the high-divergence set, smaller rank shifts (particularly among the top-2 tokens), and lower overall rates of low base-probability mass amplification. These controlled results indicate that many of the structural differences we report can indeed be attributed to the algorithmic design (eg. the clipping scheme), rather than solely to differences in training data. In addition, we have broadened both our token-level analysis and cross-sampling experiments to include a wider range of models (eg. Qwen2.5-7B Math DAPO trained with clip-high thresholds of 0.20 and 0.28, Mistral Small 24B, and models trained with SFT for comparison against RLVR), as well as a larger set of datasets (AIME-24, AIME-25, the fine-tuning dataset, and GPQA). These expanded results further reinforce the main findings reported in the original submission.

---

> ### Author Response · Authors · 2025-11-24
>
> ### (D) Control Baseline for Cross-Sampling
> > **W4:** “A random baseline is missing; this would quantify the effect of replacing high-divergence tokens.”
>
> We thank the reviewer for this helpful suggestion. We have now added a random-position cross-sampling baseline. As shown in Figure 43, randomly replacing tokens yields substantially lower accuracy than replacing high-divergence positions. While random sampling occasionally lands on some high-divergence tokens, it misses many of the crucial ones, leading to a markedly weaker effect. This further supports our conclusion that high-divergence positions carry disproportionate functional importance.
>
> ### (E) Divergence-Weighted Training (Section 6.2)
> > **W5:** “Experiment with advantage reweighting in Section 6.2 is underdeveloped…”
> >
> > **Q6:** "Have you gathered more results on advantage reweighting experiments? Currently, improvement seems very small. Have you identified for which scenarios and baseline algorithms this method is meaningfully applicable?"
>
> We thank the reviewer for this feedback. In the revised manuscript, we have expanded Section 6.2 with evaluations on additional datasets (AIME25 and AMC), where we observe consistent improvements from divergence-weighted advantages. We also clarify the motivation by connecting this intervention to the sparsity patterns, and by providing additional intuition for the high and low-kl setups.
>
> **[Table 2] Qwen2.5-Math-7B divergence-weighted Avg@32 accuracy**
>
> | Configuration | AIME-24 | AIME-25 | AMC | Overall Avg |
> | --- | --- | --- | --- | --- |
> | Baseline DAPO | 33.61 | 18.75 | 75.08 | 42.48 ± 1.35 |
> | Low-KL boost | 35.90 | 19.90 | 78.97 | 44.92 ± 0.05 |
> | High-KL boost | 36.74 | 20.00 | 78.40 | 45.05 ± 0.79 |
>
> We focus on DAPO because it is a strong baseline that already builds upon GRPO and incorporates several practical improvements. Since DAPO extends GRPO, evaluating our method on top of it may provide indication of whether our reweighting mechanism can provide any substantial differences. We also clarify that our reweighting scheme is not a KL regularization term: the KL-based weights are not included in the computation graph, and thus do not function as an explicit penalty (also, the KL in the weights need not be taken with respect to the reference model). They simply rescale the token-level advantages, making the approach compatible with algorithms, such as GRPO, that already include KL constraints.
>
> While we agree that evaluating across multiple baselines would further validate the approach, compute constraints limit our breadth of experimentation, and we therefore prioritize DAPO. Finally, we reiterate that this section is intended as an exploratory, proof-of-concept intervention designed to complement our main contributions of token-level distribution analyses and cross-sampling experiments. A more extensive study on incorporating token distributional structure into the advantages is a potential direction for future work.
>
>
>
> ### (F) Readability and Structure
> > **W6:** “The paper is dense and would benefit from moving some content to the appendix and clarifying the presentation.”
>
> We appreciate the reviewer’s suggestion. In the revised version, we have reorganized several sections, clarified the narrative flow, and moved additional technical details and secondary plots to the appendix to improve readability.
>
> ### **Responses to additional Questions**
>
> ### Q1. Entropy Visualization in Low-JS Bins
> > “In Figures 3 and 9, in the low JS bin, am I correct that RL entropy (red) is almost zero and therefore not visible? Perhaps a scatterplot or 2D histogram with JS/entropy axes would help, possibly per sequence with each point representing a token.”
>
> Thank you for the question. In the low-JS bin, the RL entropy is nearly identical to the base entropy, since low distributional divergence implies minimal entropy change. For clarity, we only plot the base entropy in this regime to avoid visually overlapping histograms; the RL entropy would be mostly overlapped with the base entropy. Our primary reason for using JS-binned plots is to clearly separate the low-divergence and high-divergence regimes, in order to analyze how they differ.
>
> We also appreciate the provided suggestion, and have added the corresponding scatter plots in Figure 22, showing JS divergence vs. entropy at the token level per sequence. These confirm our findings: a strong concentration of tokens in the low-entropy, low-JS region, and method-dependent differences in the high-JS regime (e.g. DAPO introduces additional high-divergence tokens with lower entropy).

---

> ### Author Response · Authors · 2025-11-24
>
> ### Q2. Positional Analysis Relative to Semantic Elements
> > “Have you measured position relative to system prompt, instruction, answer, etc.?”
>
> We appreciate this insightful question. We have now conducted additional experiments measuring the local divergence relative to key semantic anchors in the sequence, specifically, the model generation start and the final answer segment. The results (Figure 21) are consistent with the trends observed in the position analysis, as these elements typically occur near the beginning and end of the sequence, while we observe elevated JS divergence in the vicinity of these elements. This aligns with the intuition of making edits in initial planning or final question-answering stages when solving a problem. Qualitatively, for example, we observed that several RL-trained models consistently open their responses with stable phrases such as “To approach this problem…”, whereas the base model exhibits more variability.
>
> Importantly, however, these are not the only regions where high divergence appears. High-divergence distributions occur throughout the sequence, and our analysis focuses on identifying and characterizing some properties of these distributions, as well as demonstrating their functional importance via cross-sampling experiments. While we observe certain patterns, not all instances of high divergence fall into simple categories, and this is likely influenced by the underlying pre-training distribution of the base model.
>
> We agree that more systematically identifying such critical distributions, particularly without requiring access to the RL-trained model, would be a valuable direction for future work. This could inform more efficient RL training procedures, targeted sampling strategies, or alternative methods that achieve RL-level gains with reduced compute or data.
>
>
> ### Q3. Extending Percentage of Replaced Tokens
> > “What happens if we replace more tokens in Figure 5? Is the trend bounded by base/RL accuracy?”
>
> In general, the performance tends to reach the base or RL accuracy levels, while extending slightly beyond them (though such behavior is likely attributable to sampling noise). However, in other cases, such as Qwen2.5-32B SimpleRL on AIME25 (Figure 44a), the base with RL cross-sampling can perform substantially better than the RL model. For instance, cross-sampling base generations with only a small proportion of RL tokens can reach 16% accuracy, compared to the RL model’s <14%.
>
> The reason for this could be that the trajectories generated during cross-sampling are actually mixed trajectories: they contain mostly base-sampled tokens with only a few RL tokens injected, and although they are designed to progressively approximate RL trajectories, they are not identical, especially after only small amounts of cross-sampling steps (with subsequent tokens generated by the base model). Since the RL-generated trajectory is not guaranteed to be optimal, it is possible, as in this example, that mixed trajectories can outperform the RL outputs. This can potentially serve as a useful indicator of suboptimal RL training in certain settings.
>
>
> ### Q4. Ranking All Tokens Instead of Only High-Divergence Tokens
> > “Section 5.1 focuses on high-divergence positions; what changes if you compute the ranking metrics over all tokens? Also, please share the x-axis across columns in Figure 11(b).”
>
> Thank you for the question. The goal of Section 5 is precisely to analyze the difference between base and RL distributions at high-divergence positions, since these are where the meaningful distributional differences occur. As expected, when we compute the same ranking-based metrics over all tokens, the results show only minimal differences as the large majority of distributions lie in the low-divergence regime, where base and RL distributions are nearly identical. Aggregating over all token distributions therefore masks the very patterns Section 5 aims to highlight and makes it difficult to interpret how the high-divergence distributions actually differ.
>
> We have also followed your suggestion and updated Figure 11(b) (now it is Figure 17(b)) to use a shared x-axis, which improves comparability across columns.

---

> ### Author Response · Authors · 2025-11-24
>
> ### Q5. Entropy Shift and KL Regularization
> > “Section 3.2 about the dependency of entropy shift on position in a sequence is not substantive enough: absolute values of JS divergences shown for both methods are very low, and trend is seen only for DAPO. Also, since for DAPO the values are significantly larger than for GRPO (SimpleRL) as in section 3.1, it suggests that entropy shift might be the consequence of nonzero KL divergence regularization term and similar mechanisms to preserve the generation behaviour. Can you speculate if this is the case, or correct me if I'm wrong?”
>
> Thank you for the question. We clarify first that Section 3.2 reports JS divergence across sequence positions, not entropy. As shown in Section 3.1, distributional changes are concentrated on a small subset of tokens, so when averaging divergence across entire sequences, the means of the sequential JS values are expectedly small for both methods. The purpose of Section 3.2 is therefore to examine whether there are sequence regions where divergence consistently concentrates.
>
> Regarding the reviewer’s hypothesis about whether the observed trends may arise from the KL regularization or similar mechanisms: this is an interesting point. To disentangle this, we include additional results on Qwen2.5-Math 7B where we compare upper clip 0.2 vs. 0.28, both without any KL regularization. Even in this setting, we observe differences in divergence patterns and entropy differences that align with the main results. Specifically regarding entropy shift, Figure 33 compares the 0.2 and 0.28 upper clip, where the 0.2 DAPO model exhibits significantly stronger entropy collapse among its divergent distributions, with very few low-base-entropy distributions with high-divergence (similar to that of the SimpleRL model, which also uses an upper clip of 0.2). This suggests that the entropy shift is not solely a consequence of KL divergence. However, the KL regularization indeed appears to enforce even further sparsity of divergence, as expected.
>
> Finally, we thank the reviewer for suggesting [2], which we have now included in the Related Work section.
>
> We also thank the reviewer for the thorough critique and welcome any further discussion.
>
> [2] Reasoning with Sampling: Your Base Model is Smarter Than You Think, Karan A., Du Y., 2025.

---

### Official Review · Reviewer_pdCB · 2025-11-01

**Soundness:** 4
**Presentation:** 3
**Contribution:** 3
**Rating:** 6
**Confidence:** 3

**Summary:**

This paper examines how token-level distributional properties shift during RLVR fine-tuning and explores the functional significance of these changes.

Empirically, the authors find that RL fine-tuning induces distributional divergence in only a small subset of tokens, mostly occurring early in the fine-tuning process. Notably, the magnitude of this divergence is not fully predicted by token entropy.

To assess the functional role of these divergent RL-sampled tokens, the authors inserted some of them into generations produced by the base model. They observed that this intervention caused the base model to perform as well as the RL fine-tuned model. Conversely, as expected, inserting base model tokens into the RL model’s outputs reduced its performance to base model levels.

Furthermore, the authors demonstrate that modifying GRPO/DAPO to weigh these divergent tokens differently—rather than treating all tokens uniformly—during fine-tuning results in improved downstream performance. This highlights a practical value for these divergent tokens.

**Strengths:**

I found the article to be generally well written. The motivation behind each analysis was clear, and the findings were explained well. The authors’ analyses were thorough—they not only examined how token distributions change, but also investigated which factors might predict these changes, the functional role of the divergent tokens, and how these insights can be leveraged to improve performance of RL fine-tuning.

**Weaknesses:**

It remains unclear under which specific sequences of tokens these observations were made, and clarifying this would enhance the paper. Additionally, the motivation for using JS divergence over KL divergence could be explained more thoroughly—currently, it is addressed in only a couple of sentences, but this section could be expanded with a simple illustrative example. Especially since the rest of the paper's relies on this observation.

To streamline the narrative, consider moving certain experiments, such as the analysis of top-k token overlap, to the appendix, as these largely reinforce points already demonstrated by earlier results. At present, the paper contains too many small results, some of which make similar points using different approaches. This redundancy could be reduced; focusing on a few key analyses, with detailed motivation and deeper exploration of findings, would strengthen the paper.

Doing so, could also free up space for the final analysis, which felt somewhat crammed. I believe this final result was particularly important and deserved more attention than it currently receives.

**Questions:**

It would be good to verify whether these observations hold not only for the Qwen model family, but also extend to other open-source model families such as Llama or Mistral families.

I am also still unclear about why altering the placement of such a small number of tokens after RL fine-tuning can have such a strong impact on downstream performance. Is this function of the task/context within which these analyses were conducted as that is still unclear to me. Could the authors elaborate on their intuition for why this effect occurs?

---

> ### Author Response · Authors · 2025-11-24
>
> We thank the reviewer for the thoughtful feedback and positive assessment. Below we discuss and address the points raised by the reviewer.
>
> ### (A) Clarification of Which Sequences Are Analyzed & Why
> > “It remains unclear under which specific sequences of tokens these observations were made, and clarifying this would enhance the paper.”
>
> Thank you for bringing this up. In the revision, we now explicitly describe which sequences are used in each analysis:
> For the token-level divergence analysis, we evaluate divergences primarily on RL-generated responses, as we treat the RL trajectory as the target trajectory whose deviations from the base model we aim to understand. The motivating question for this perspective is how do we get from our base model to the RL-trained model?
>
> - For the cross-sampling experiments, the sequences are inherently mixed trajectories, since we insert tokens from one model into the other’s rollout.
> - In forward cross-sampling, we generate with the base model, and insert a small proportion of RL tokens at high-divergence positions, gradually “approximating” the RL reasoning path.
> - In reverse cross-sampling, we instead begin with RL and insert a small proportion of base tokens, approximating a regression toward the base model.
>
>
> ### (B) Expanded Motivation for JS Divergence vs. KL Divergence
> > “Additionally, the motivation for using JS divergence over KL divergence could be explained more thoroughly—currently, it is addressed in only a couple of sentences, but this section could be expanded with a simple illustrative example. Especially since the rest of the paper relies on this observation.”
>
> We appreciate the comment, and have expanded the motivation in the revised version. Our goal is to quantify how “different” the token-level distributions of the base and RL models are. While many probability divergences could in principle be used, we choose JS divergence because it offers several practical and methodological advantages for our empirical analysis. First, JS divergence is bounded, providing a normalized and interpretable scale. This avoids the unboundedness of KL divergence, which can disproportionately amplify small probability differences and obscure the underlying structural trends we aim to study. In addition, KL divergence becomes undefined when the support of one distribution is not contained in the other. Although softmax distributions assign nonzero probability to all vocabulary tokens, in practice large-scale inference via vLLM does not retrieve probabilities for the entire vocabulary at each step. KL divergence may therefore be unstable in this truncated-support setting. Similarly, JS divergence remains well defined under top-p truncation of the distributions (which is typically used during inference).
>
>
>
> ### (C) Streamlining Redundant Experiments
> > “To streamline the narrative, consider moving certain experiments, such as the analysis of top-k token overlap, to the appendix, as these largely reinforce points already demonstrated by earlier results…”
>
> We appreciate this suggestion, and have moved some of the analysis and several smaller supporting plots to the appendix, reducing
>  redundancy in the main text. We would also like to clarify that while earlier sections of the paper quantified the extent and spread of these distributional changes, as well as their relationships with entropy regimes, it did not reveal the detailed mechanics of, how probability mass is redistributed within these critical high-divergence positions, which is what Section 5 aims to study.
>
>
> ### (D) Expanded Final Analysis Section
> > “The final analysis felt somewhat crammed… this result was particularly important and deserved more attention.”
>
> Thank you for pointing this out. In the revised version, we have expanded
> this section to provide additional motivation, clearer framing, and more
> thorough discussion. Specifically, we now elaborate on and
> include:
>  - how the approach connects to the sparsity and other findings established earlier in the paper,
>  - the intuition behind weighting high- vs. low-divergence tokens,
>  - Evaluation across additional datasets (AIME25 and AMC)
>
> This expanded discussion clarifies the purpose of the divergence-weighted experiment as an exploratory demonstration as a proof-of-concept informed by our token-level divergence analysis.

---

> ### Author Response · Authors · 2025-11-24
>
> ### (E) Generalization Beyond Qwen
> > “Do these observations hold not only for the Qwen family, but also for other model families such as Llama or Mistral?”
>
> Thank you for bringing this up. Yes. In the revision, we now include token-level divergence analysis for Mistral Small 24B, trained with SimpleRL, and observe the same sparsity patterns and consistent cross-sampling behavior. In addition, we have broadened both our token-level analysis and cross-sampling experiments to include a wider range of models (eg. Qwen2.5-7B Math DAPO trained with clip-high thresholds of 0.20 and 0.28, Mistral Small 24B, and models trained with SFT for comparison against RLVR), as well as a larger set of datasets (AIME-24, AIME-25, the fine-tuning dataset, and GPQA). These expanded results further reinforce the main findings reported in the original submission.
>
> ### (F) Why Small Token Changes Have Large Downstream Impact
> > “Why does altering a small number of tokens after RL fine-tuning have such a strong impact on downstream performance? Could you elaborate on the intuition for this effect?”
>
> We thank the reviewer for highlighting this point, and we agree that the cross-sampling results initially appear surprising. We clarify a few key points:
> - We are not exactly altering the placement of tokens; we are replacing a small number of tokens with those from the other model. These substitutions occur at positions where the base and RL policies differ significantly.
> - From our Section 3 analyses, conditioned on RL-generated sequences, only a small portion of tokens exhibit meaningful distributional differences. These represent the positions where RL most strongly modifies the reasoning trajectory.
>
> **Forward cross-sampling intuition (base → RL):**
> When generating tokens with the base model, if we were to, at all non-zero divergence token distributions, inject an RL-sampled one, the resulting sequence would closely approximate the RL trajectory, and thus we may expect RL-level performance. Based on our previous analysis, we would also expect the amount of RL-sampled tokens required to be relatively small. So, by performing the cross-sampling process, we are progressively getting closer to an "approximation" of the RL trajectory.
>
> What is more interesting, and now highlighted more clearly, is that we find that even replacing just the first few high-divergence tokens and then completing the response with the base model produces noticeable accuracy gains, and the gains progress consistently as more RL tokens are inserted. This indicates that, on average, each high-divergence token distribution contributes a meaningful improvement, that even the base model can obtain just by continuing to generate from such a mixed trajectory.
>
> **Reverse cross-sampling intuition (RL → base):**
> Similarly, replacing RL tokens with base tokens at high-divergence positions leads to consistent degradation of performance. Even when replacing only a small fraction of tokens and then letting the RL model complete the sequence, the answer quality declines (the RL model isn't able to fully "recover" from the base-introduced tokens). This provides evidence that these positions are functionally important.
>
> **Overall interpretation:**
> The near-monotonic curves, rather than sporadic jumps, demonstrate that high-divergence tokens encode accumulated, structurally meaningful modifications introduced by RL. We hypothesize that these critical distributions likely dependent the pre-training data, and that RL modifies token distributions that can make it more likely for the base model to complete a correct solution.
>
> We appreciate your insights throughout and are glad to answer any additional questions that may come up.

---

### Official Review · Reviewer_3wey · 2025-11-01

**Soundness:** 2
**Presentation:** 3
**Contribution:** 3
**Rating:** 4
**Confidence:** 4

**Summary:**

This paper investigates how Reinforcement Learning with Verifiable Rewards (RLVR) fine-tuning affects large language models (LLMs) at the token level. While RLVR is known to significantly improve LLM reasoning performance, the paper demonstrates that these improvements are not distributed evenly across outputs. Instead, RLVR induces sparse and highly targeted changes: only a small fraction of tokens in model responses are significantly altered by RL. Notably, the paper finds that these high-divergence token changes are not restricted to regions of high model uncertainty, but often override even confident base predictions. The authors conduct detailed empirical analyses to characterize these changes, evaluate their functional importance, and propose new RL objective variants to leverage these insights.

**Strengths:**

- The paper provides a novel token-level analysis of distributional shifts caused by RLVR, revealing sparsity and context-sensitivity in model updates that are not addressed by aggregate or entropy-based approaches in previous work. The authors introduce creative cross-sampling experiments that directly test the functional impact of high-divergence tokens and propose divergence-weighted RL objectives to exploit the observed sparsity.

- The empirical methodology is thorough and rigorous, comprising comprehensive token-level and sequence-level analyses, comparisons across several RLVR algorithms and model scales, and carefully designed interventions. Quantitative findings are supported by clear metrics (JSD, accuracy, percentage of affected tokens) and transparent reporting of results.

- The paper is well-organized and clearly written, with all key concepts (e.g., divergence, entropy, cross-sampling) explicitly defined and illustrated. Figures and tables are used effectively to present the main findings, and the logical structure guides the reader through the motivation, experiments, results, and implications.

- The work offers new insights into the mechanisms of RLVR, showing that performance gains stem from a small set of critical token-level changes. Its findings have practical implications for designing more efficient, interpretable, and targeted RLVR fine-tuning strategies for LLMs.

**Weaknesses:**

1. **Narrow dataset scope and lack of experimental details.** The experiments focus exclusively on math reasoning (AIME24) with Qwen2.5 models, so it is unclear whether findings about token-level sparsity and cross-sampling generalize to other domains, reasoning tasks, and models. The paper uses a fixed sampling setup (32 samples/problem, top-p=0.7, temperature=1), but omits ablations on these choices. No statistical significance or variance is reported for the divergence-weighted advantage gains in Table 2. JS divergence is calculated on top-p–truncated distributions, which can distort divergence magnitudes, yet the paper provides no discussion of this issue. The experimental setup is described at a high level, but several key implementation details necessary for full reproducibility (e.g., cross-sampling thresholds) are omitted.

2. **Limited causal and interpretability analysis.** While cross-sampling demonstrates the criticality of high-divergence tokens, the paper does not deeply investigate why these tokens emerge, what properties of inputs make them critical, or how specific token changes affect reasoning trajectories and outcomes.

3. **On-policy conditioning bias.** Token divergences are measured along RL trajectories only, which may bias where "critical" tokens appear.

**Questions:**

1. Can you provide results or analyses on non-math tasks or with other LLM architectures to support generalization claims?
2. How sensitive are your main findings to the sampling parameters (top-p, temperature, number of samples)? Have you run any ablations?
3. Can you report statistical significance or variance for the divergence-weighted objective gains in Table 2? Are these improvements robust across seeds or runs?
4. How does using JS divergence on top-p–truncated distributions affect your results? Can you show that your findings are not artifacts of truncation?
5. What exact divergence thresholds and implementation details were used in cross-sampling experiments? Can you provide all necessary information for reproducibility?
6. Have you analyzed what input features or contexts make high-divergence tokens critical? Any qualitative or quantitative results on the causes?
7. Could measuring divergences on off-policy or mixed trajectories affect the identification of critical tokens? Have you considered or tested for on-policy bias?

---

> ### Author Response · Authors · 2025-11-24
>
> We thank the reviewer for their detailed comments and constructive feedback. Below, we address the questions/concerns raised by the reviewer. All revisions referenced here have already been incorporated into the updated manuscript.
>
> ### (A) Broader Evaluation Scope Across Models, Tasks, and Training Setups
> > “Narrow dataset scope and lack of experimental details. The experiments focus exclusively on math reasoning (AIME24) with Qwen2.5 models, so it is unclear whether findings about token-level sparsity and cross-sampling generalize to other domains, reasoning tasks, and models.”
> >
> > “Can you provide results or analyses on non-math tasks or with other LLM architectures to support generalization claims?”
>
> We appreciate the reviewer’s comment and agree that evaluating on more datasets and model families is important for assessing generality. In the revised manuscript, we substantially broaden both the model roster (Qwen2.5-7B Math DAPO with clip-high values 0.20/0.28, Mistral Small 24B, and SFT-trained models to compare with RLVR) and the task suite (AIME-24/25, the DAPO fine-tuning set, AMC, and GPQA). These additions reinforce the original findings: distributional shifts remain sparse, high-divergence positions stay functionally critical in cross-sampling, and Section 6 now reports divergence-weighted intervention results on AIME-25 and AMC.
>
> ### (B) Sampling Hyperparameters and Ablations
> > “The paper uses a fixed sampling setup (32 samples/problem, top-p=0.7, temperature=1), but omits ablations on these choices.”
> >
> > “How sensitive are your main findings to the sampling parameters (top-p, temperature, number of samples)? Have you run any ablations?”
>
> Our default sampling parameters follow common practice for DAPO-trained Qwen models (top-p=0.7, temperature=1), and we keep them for consistency across models/configurations. To test robustness, we added ablations varying the sampling setup (e.g., top-p = 0.8 and 0.9). The token-level findings remain stable, as summarized in Appendix A.4.1 (Figures 25–26).
>
>
> ### (C) Statistical Significance and Variance for Divergence-Weighted Training
> > “No statistical significance or variance is reported for the divergence-weighted advantage gains in Table 2.”
> >
> > “Can you report statistical significance or variance…? Are these improvements robust across seeds or runs?”
>
> We now include standard deviations for average accuracy in Table 2, computed across multiple runs. Divergence-weighted training shows consistent improvements, though the baseline itself exhibits some run-to-run noise.
>
> ### (D) JS Divergence Under Top-p Truncation
> > “JS divergence is calculated on top-p–truncated distributions, which can distort divergence magnitudes, yet the paper provides no discussion of this issue.”
> >
> > “How does using JS divergence on top-p–truncated distributions affect your results? Can you show that findings are not artifacts of truncation?”
>
> We used truncated distributions to reflect the effective sampling distribution, but now also compute JS on the full (non-truncated) distributions. Figure 26 shows that while divergence values increase slightly, sparsity patterns persist, confirming the findings are not artifacts of truncation.
>
> ### (E) Reproducibility and Implementation Details
> > “The experimental setup is described at a high level, but several key implementation details necessary for full reproducibility (e.g., cross-sampling thresholds) are omitted.”
> >
> > “What exact divergence thresholds and implementation details were used in cross-sampling…?”
>
> We appreciate the reviewer bringing this up. Appendix A.3.2 now includes the cross-sampling thresholds for the cross-sampling experiments, while Algorithm 1 summarizes the procedure. Appendix A.3.3 details the training hyperparameters, data sources, and sampling setups.

---

> ### Author Response · Authors · 2025-11-24
>
> ### (F) What makes high-divergence tokens critical & Additional Interpretability Analyses
> > “Limited causal and interpretability analysis… the paper does not deeply investigate why these tokens emerge, what properties of inputs make them critical, or how specific token changes affect reasoning trajectories.”
> >
> > “Have you analyzed what input features or contexts make high-divergence tokens critical?”
>
> The reviewer brings up a good point on additional interpretability analyses. Section 3.4 shows that high-divergence tokens often include transitional words, reasoning phrases, and equation fragments, while low-divergence tokens tend to be numerals/operators. However, many tokens that are commonly in high-divergence distributions are not always in the high-divergence regime, and so token semantic identity is not enough to characterize these distributions.
>
> We have also conducted additional experiments measuring the local divergence relative to key semantic anchors in the sequence, specifically, the model generation start and the final answer segment. The results (Figure 21) are consistent with the trends observed in the position analysis. Because these elements typically occur near the beginning and end of the sequence, we observe elevated JS divergence in their vicinity. This matches the intuition of making edits in initial planning or final question-answering stages. Qualitatively, for example, several RL-trained models consistently open their responses with stable phrases such as “To approach this problem…”, whereas the base model exhibits more variability in their openings.
>
> Importantly, however, these are not the only regions where high divergence appears. High-divergence tokens occur throughout the sequence, and our analysis primarily focuses on identifying and characterizing some properties of these distributions, as well as demonstrating their functional importance via causal cross-sampling experiments. While we observe certain patterns, not all instances of high divergence fall into simple categories, and this is likely influenced by the underlying pre-training distribution of the base model.
>
> We agree that more systematically identifying such critical distributions, particularly without requiring access to the RL-trained model, would be a valuable direction for future work. This could inform more efficient RL training procedures, targeted sampling strategies, or alternative methods that achieve RL-level gains with reduced compute or data.
>
> ### (G) On-Policy Conditioning Bias
>  > “Token divergences are measured along RL trajectories only, which may bias where ‘critical’ tokens appear.”
>  > “Could measuring divergences on off-policy or mixed trajectories affect identification of critical tokens?”
>
> We appreciate the reviewer’s question regarding potential on-policy bias. Our primary
> analysis computes divergences along RL-generated trajectories, since we treat the RL
> rollout as the target trajectory whose deviations from the base model we aim to
> understand. This motivates our forward cross-sampling experiments, as we effectively
> “approximate” RL trajectories using the base-generated trajectories, but with small
> amounts of RL-sampled tokens. On the other hand, measuring divergences on responses
> produced by the base model is related to our reverse cross-sampling experiment, as
> the RL trajectories are being mixed with small amounts of base-sampled tokens to
> “approximate” the base trajectories.
>
> Moreover, our cross-sampling experiments already operate on mixed trajectories, since
> they replace small amounts of tokens from the other model during the generation
> process. The fact that both forward and reverse cross-sampling produce progressive
> changes in accuracy, while requiring only a small portion to recover the other
> model’s performance, further indicates that the identification of critical tokens may
> not be an artifact of purely on-policy evaluation.
>
> Thank you again for the detailed feedback; please let us know if any further clarifications would be useful.

---

> > ### Comment · Reviewer_3wey · 2025-11-26
> >
> > Thank you for the clarifications. The expanded evaluation to more models and tasks, the added ablations, and the clearer reporting address the main concerns. Your results convincingly show that RLVR improvements are driven by sparse, high-divergence tokens, with strong evidence from both cross-sampling experiments and divergence-weighted objectives. However, the scope remains focused primarily on reasoning tasks, and the interpretability analysis could go deeper. Therefore, I am raising my score to 6.

---

### Official Review · Reviewer_MBtv · 2025-11-06

**Soundness:** 3
**Presentation:** 2
**Contribution:** 2
**Rating:** 4
**Confidence:** 3

**Summary:**

This paper analyzes how Reinforcement Learning with Verifiable Rewards (RLVR) modifies token-level output distributions during RL-tuning. The authors first examine distributional shifts between base and RL-finetuned models using Jensen-Shannon divergence, finding that RL-induced changes are highly sparse, with only a small fraction of tokens exhibiting significant divergence. Furthermore, through the Cross-Sampling experiments, they confirm that the effect of RL-tuning stems from the reranking of high-divergence tokens. Motivated by these findings, the authors propose a divergence-weighted advantage modification to the RL objective, showing potential improvement against existing RL-tuning.

**Strengths:**

1. The structure of this paper is clear; the authors first provide a thorough analysis of the token-level distribution shift and highlight the importance of the high-divergence tokens during RL fine-tuning. They then proposed an improved divergence-weighted objective based on their findings, making the paper sound and logical.
2. The analysis of token-level divergence and corresponding ablations is insightful and interesting. Although empirical, they are critical in understanding the operation of RL fine-tuning methods from the token-level aspect.

**Weaknesses:**

1. It is suggested to proofread the paper, especially the experiments. For example, in Table 2, the configuration details should be presented more clearly.
2. The proposed method is evaluated in a rather limited setting; for instance, it is tested only on the AIME-2024 dataset. Moreover, the experiments lack comparisons with related works [1].
3. The improvement achieved by the proposed method appears modest. According to the results in Table 2, the gain in Avg@32 is limited.

[1] Beyond the 80/20 rule: High-entropy minority tokens drive effective reinforcement learning for llm reasoning

**Questions:**

1. What is the performance of the proposed method against other benchmarks?
2. In Table 2, what is the pass@1 performance?
3. What is the difference between emphasizing high-divergence and low-divergence during RL fine-tuning? In Table 2, both of these methods show similar performance gain. Could you explain this?

---

> ### Author Response · Authors · 2025-11-24
>
> We thank the reviewer for their careful reading and constructive feedback. Below we address the main concerns, and all corresponding revisions have been incorporated into the updated manuscript.
>
> ### (A) Clarity of Experimental Details
> > “It is suggested to proofread the paper, especially the experiments. For example, in Table 2, the configuration details should be presented more clearly.”
>
> We appreciate the feedback and suggestions. The experimental section has been restructured for readability: improving the clarity and organization of the experimental sections, providing additional details on the token analysis, cross-sampling experiments, and divergence-weighted training. The complete set of sampling hyperparameters, training details, and cross-sampling thresholds are detailed in Appendix A.3, which we reference explicitly in the main text. Figure captions and the remaining tables were also proofread for consistent terminology.
>
> ### (B) Limited Evaluation Scope & Comparison to Related Work [1]
> > “The proposed method is evaluated in a rather limited setting; for instance, it is tested only on the AIME-2024 dataset. Moreover, the experiments lack comparisons with related works [1].”
> >
> > “What is the performance of the proposed method against other benchmarks?”
>
> We agree with the reviewer that broader evaluation would be valuable. Section 6 now reports divergence-weighted advantage experiments on AIME-24, AIME-25, and AMC, demonstrating consistent improvements over the baseline DAPO (Table 2). We additionally integrate the entropy-based clipping strategy from [1] into our pipeline and report its results in Table 5. Our divergence-guided intervention outperforms the entropy-clip baseline in the settings we test. As noted in [1], their method mainly produced performance gains for much larger models (e.g., 32B), whereas such large-scale RLVR training is beyond our compute budget.
>
> **Divergence-weighted Avg@32 accuracy**
>
> | Model / Configuration | Source Table | AIME-24 | AIME-25 | AMC | Overall Avg |
> | --- | --- | --- | --- | --- | --- |
> | Qwen2.5-Math-7B Baseline DAPO | Table 2 | 33.61 | 18.75 | 75.08 | 42.48 |
> | Qwen2.5-Math-7B Low-KL boost | Table 2 | 35.90 | 19.90 | 78.97 | 44.92 |
> | Qwen2.5-Math-7B High-KL boost | Table 2 | 36.74 | 20.00 | 78.40 | 45.05 |
> | Qwen2.5-Math-7B 80/20 clip entropy | Table 5 | 35.26 | 17.03 | 72.68 | 41.66 |
>
> Beyond the intervention study, Sections 3–5 and Appendix A.4 now cover a wider range of models (eg. Qwen2.5-7B Math DAPO trained with clip-high thresholds of 0.20 and 0.28, Mistral Small 24B, and models trained with SFT for comparison against RLVR), as well as a larger set of datasets (AIME-24, AIME-25, the fine-tuning dataset, and GPQA). These expanded results further reinforce the main findings reported in the original submission.
>
> [1] Beyond the 80/20 rule: High-entropy minority tokens drive effective reinforcement learning for llm reasoning
>
> ### (C) Magnitude and Interpretation of Improvements (Table 2)
> > “The improvement achieved by the proposed method appears modest. According to the results in Table 2, the gain in Avg@32 is limited.”
> >
> > “In Table 2, what is the pass@1 performance?”
>
> Our reported Avg@32 metric is the standard pass@1 estimator, computed using 32 samples per problem following the pass@k formulation. We have clarified this explicitly in the table caption to avoid confusion.
> Regarding the magnitude of the gains: we agree that the improvements from the reweighting strategy in Section 6 may seem modest. However, the intervention involves just a small reweighting to the advantages, without fundamental algorithmic differences, yet yields measurable improvements across multiple datasets. More importantly, this reweighting component is presented as an exploratory proof-of-concept informed by our token-level divergence analysis. The core contributions of the paper remain the token-level distributional comparisons and the cross-sampling causal experiments, which reveal a sparse set of influential tokens and characterize how RL alters reasoning trajectories and token distributions.

---

> ### Author Response · Authors · 2025-11-24
>
> ### (D) High- vs Low-Divergence Weighting Behavior
> > “What is the difference between emphasizing high-divergence and low-divergence during RL fine-tuning? In Table 2, both of these methods show similar performance gain. Could you explain this?”
>
> This is a very insightful question. The two divergence-weighting strategies are motivated by distinct observations from our earlier analyses:
>
> - High-divergence emphasis reflects the fact that RL induces concentrated changes at key positions that causally influence the reasoning path (as demonstrated via cross-sampling). Emphasizing these positions may reinforce the distributional modifications that RL has already introduced at each PPO step.
> - Low-divergence emphasis explores the complementary hypothesis that RL may over-focus on a very small set of tokens. Giving slightly more weight to stable/low-divergence positions may broaden the effective support of updates and potentially improve robustness.
>
> Since these are conceptually distinct hypotheses, we evaluated both. The similar performance aligns with our broader takeaway that Section 6 should be interpreted as exploratory, illustrating how token-level divergence signals may inform training design.
>
> We again appreciate your thoughtful questions and feedback and would be happy to clarify anything further if additional concerns arise.

---

### Author Response · Authors · 2025-12-04
**Rebuttal Summary**

Dear AC and reviewers,

Thank you for overseeing and putting your time and efforts into the review process. We provide below a consolidated summary of the main strengths highlighted by the reviewers and how our revision directly addresses the weaknesses/questions raised. We hope this helps contextualize the changes and clarifications made in the updated manuscript and rebuttal responses.

---

## **Strengths Identified by Reviewers**

**Clarity and Organization (Reviewers MBtv, 3wey, pdCB)**
Multiple reviewers noted that the paper is clearly written, well organized, and presents a logical flow from motivation to analysis to intervention. Key concepts such as divergence, entropy, and cross-sampling are clearly defined and supported by intuitive motivation.

**Novel and Insightful Token-Level Analysis (Reviewers MBtv, 3wey, pdCB, SHpG)**
Reviewers agreed that the token-level divergence analysis is new and sheds meaningful light on how RLVR alters model behavior. They specifically emphasized the discovery of sparsity in distributional shifts and the identification of a small set of critical token positions.

**Cross-Sampling Experiments as a Functional Probe (Reviewers 3wey, SHpG)**
Reviewers found the cross-sampling methodology clever and impactful. These experiments directly demonstrate that high-divergence tokens are functionally important and sufficient to recover or degrade RL-level reasoning.

**Thorough Methodology and Strong Empirical Rigor (Reviewers 3wey, pdCB)**
Reviewers highlighted the comprehensive experimental setup, use of multiple quantitative metrics, and detailed sequence- and token-level evaluations.

---

## **How the Revision Addresses Reviewer Concerns**

Below is a reviewer-by-reviewer summary of the major concerns and how they were addressed in the revised manuscript and rebuttal responses.

**Reviewer MBtv**

**Concerns:**
1. Lack of clarity in experimental details and table captions
2. Limited evaluation on AIME-24 only; also wanted comparison with [1]
3. Modest magnitude of improvement in Table 2
4. Clarification of differences between high-divergence and low-divergence weighting

**Revisions Made:**
- The experimental section was restructured for clarity, and all sampling hyperparameters, training configurations, and cross-sampling thresholds are now given explicitly in Appendix A.3.
- Evaluation on training intervention expanded to AIME-25 and AMC, with consistent improvements reported.
- Additional explanation on the conceptual differences between high- and low-divergence weighting strategies was added, with expanded discussion in Section 6.
- Added RLVR training results from entropy-clip method from [1].

[1] Beyond the 80/20 rule: High-entropy minority tokens drive effective reinforcement learning for llm reasoning

---

**Reviewer 3wey**

**Concerns:**
1. Narrow scope of models and tasks
2. Missing ablations on sampling hyperparameters
3. Variance and robustness of divergence-weighted training
4. Effect of top-p truncation on JS divergence
5. Need for more interpretability analysis
6. Potential bias from evaluating only RL trajectories
7. Missing implementation details

**Revisions Made:**
- Added models: Mistral Small 24B (SimpleRL), Qwen2.5 32B SFT model, and multiple Qwen2.5 7B-Math DAPO variants (comparing upper clips of 0.28 and 0.2).
- Added additional datasets: AIME-25, AMC, GPQA, and the fine-tuning dataset.
- Added sampling ablations (top-p = 0.8 and 0.9) showing stability.
- Added standard deviations for divergence-weighted training results.
- Recomputed JS divergence on full distributions to show sparsity persists beyond top-p truncation.
- Added additional semantic and positional analysis of high-divergence tokens.
- Clarified on-policy and mixed-policy evaluation; in our token-distribution analysis, the RL trajectories are used as they are treated as a "target trajectory", while cross-sampling inherently uses mixed trajectories.
- Expanded Appendix A.3 with complete implementation details.

---

> ### Author Response · Authors · 2025-12-04
>
> ---
>
> **Reviewer pdCB**
>
> **Concerns:**
> 1. Clarification of which sequences are analyzed and why
> 2. More detailed justification for using JS divergence over KL
> 3. Some analyses could be moved to appendix
> 4. Final section felt compressed
> 5. Generalization beyond Qwen
> 6. Intuition for why small token changes have large effects
>
> **Revisions Made:**
> - Clarified which sequences are used in each analysis (RL trajectories for token-distribution analysis, mixed trajectories for cross-sampling).
> - Expanded motivation for JS divergence for the analysis, including stability under truncation and advantages over KL.
> - Moved some plots and supporting analyses to the appendix to streamline narrative.
> - Expanded the divergence-weighted training section with clearer framing and added results on AIME-25 and AMC.
> - Added additional results for Mistral Small 24B and SFT models.
> - Added further intuition discussing why small sets of substituted tokens produce large functional changes.
>
> ---
>
> **Reviewer SHpG**
>
> **Concerns:**
> 1. Limited treatment of other post-training methods (SFT, preference optimization, RLHF)
> 2. Limited generality beyond the Qwen family and math datasets
> 3. Potential dataset confounds in GRPO vs DAPO comparisons
> 4. Missing random baseline for cross-sampling
> 5. Advantage reweighting experiment underdeveloped
> 6. Readability and structure could be improved
> 7. Detailed questions regarding entropy visualization, positional analysis, extensions of cross-sampling, and ranking metrics
>
> **Revisions Made:**
> - Added SFT experiments on token-distribution analysis using Qwen2.5-32B, showing noticeably different structural behavior from RLVR.
> - Added analyses on Mistral Small 24B and GPQA to support generality across model families and tasks.
> - Added controlled DAPO experiments with clip-high values of 0.20 and 0.28 trained on the same dataset to isolate algorithmic effects.
> - Added a random-position cross-sampling baseline, which performs substantially worse than divergence-based cross-sampling under the same amount of token swaps, highlighting the importance of high-divergence token distributions.
> - Expanded Section 6.2, added evaluation on additional datasets (AIME25 and AMC), and clarified the purpose and limitations of divergence-weighted training.
> - Improved readability by moving some additional technical content to the appendix and reworking figure layouts.
> - Added new figures and explanations addressing the reviewer’s specific technical questions.
>
> ---
>
> ## **Final Remarks**
>
> We appreciate the reviewers’ thoughtful feedback, which led to a revised version that significantly strengthens the paper by broadening the evaluations, clarifying methodological details, and adding additional analyses that directly address their questions. The core findings, particularly the sparsity of RLVR-induced token-level shifts and their causal significance demonstrated via cross-sampling (as well as the more fine-grained distributional analysis of how the token-level distributions change), remain strongly supported and consistently occur across model families and training setups.
>
> Thank you again for your time and consideration.

---

### Meta-Review · Area_Chair_53KT · 2026-01-07

**Summary:**

This paper gives an interesting empirical analysis of a sparsity phenomenon that happens during RLVR fine-tuning. Based on that phenomenon, some practical methods are proposed, but the primary contribution is the experimental study of the behavior. While reviewers had some initial very-valid concerns, I think these have been adequately addressed in the rebuttal, and given the quality of the analysis and the timeliness of the topic, am happy to recommend acceptance of this paper for this ICLR.



It is worth noting that this submission is closely related to the separate, parallel submission https://openreview.net/forum?id=2OO399hRD6. In the final version, you should cite this paper and discuss the differences.

**Reviewer Concerns:**

Several reviewers expressed concerns about limitations of the analysis, particularly (in the initial version) to a single model family on a single dataset. The rebuttal added experiments with several model families on several datasets, showing consistency. Other questions, like why use Jensen-Shannon rather than KL, were also fully addressed to my satisfaction.

**Reviewer Scores:**

MBtv did explicitly upgrade their score in a comment (4 to 6). Other concerns were all, in my opinion, adequately addressed by the rebuttal, and hence other reviewers' scores could have gone up as well.

---

### Decision · Program_Chairs · 2026-01-26

Accept (Poster)